# The Field Monitoring Experiment of the Roof Strata Movement in Coal Mining Based on DFOS

**DOI:** 10.3390/s20051318

**Published:** 2020-02-28

**Authors:** Tao Hu, Gongyu Hou, Zixiang Li

**Affiliations:** Mechanics and Civil Engineering, China University of Mining and Technology, Beijing 100083, China; hgyht@cumtb.edu.cn (G.H.); BQT1900605043@student.cumtb.edu.cn (Z.L.)

**Keywords:** roof strata movement, distributed optical fiber sensing technology, optical fiber sensing performance, optical fiber strain transfer performance, anchorage coupling experiment

## Abstract

Mining deformation of roof strata is the main cause of methane explosion, water inrush, and roof collapse accidents amid underground coal mining. To ensure the safety of coal mining, the distributed optical fiber sensor (DFOS) technology has been applied in the 150,313 working face by Yinying Coal Mine in Shanxi Province, north China to monitor the roof strata movement, so as to grasp the movement law of roof strata and make it serve for production. The optical fibers are laid out in the holes drilled through the overlying strata on the roadway roof and BOTDR technique is utilized to carry out the on-site monitoring. Prior to the on-site test, the coupling test of the fiber strain in the concrete anchorage, the calibration test of the fiber strain coefficient of the 5-mm steel strand (SS) fiber, and the test of the strain transfer performance of the SS fiber were carried out in the laboratory. The approaches for fiber laying-out in the holes and fiber’s spatial positioning underground the coal mine have been optimized in the field. The indoor test results show that the high-strength SS optical fiber has a high strain transfer performance, which can be coupled with the concrete anchor with uniform deformation. This demonstrated the feasibility of SS fiber for monitoring strata movement theoretically and experimentally; and the law of roof strata fracturing and collapse is obtained from the field test results. This paper is a trial to study the whole process of dynamic movement of the deformation of roof strata. Eventually the study results will help Yinying Coal Mine to optimize mining design, prevent coal mine accidents, and provide detailed test basis for DFOS monitoring technique of roof strata movement.

## 1. Introduction

The deformation of roof strata caused by mining is the main cause of varieties of coal mining accidents including methane accumulation, roof water inrush, rock collapse, and other accidents. The monitoring and measurement of movement of roof strata is the key for gas drainage, prevention for water inrush, and other roof accident. The strata movement caused by coal mining is a “black box.” At present, it is only explained in the control principle, reaching the “gray box” level [1]. In the exploration and research of “black box” to “gray box” of mining strata movement, numerous hypothesis have been attempted worldwide that have been guiding production in practice, such as Academician Qian Minggao’s “Masonry Beam Theory” (MBT) [2] and “Key Layer Theory (KLT)” [3], Academician He Manchao’s “Cutting Cantilever Beam Theory” (CCBT) [4], etc. Based on Qian’s theory: With the continuous advance of the working face, the goaf becomes larger and larger, and the top plate area of the vertical decompression area increases. Because of the combined action of ground pressure and water pressure in the vertical direction, the top plate, as a bearing structure, will reach its strength limit. First, cracks appear in the middle of the long axis (at the maximum bending moment) and propagate to both ends along the long axis direction; when the two cracks increase to a certain length, cracks appear in the middle of the short axis and propagate to both ends; second, the cracks continue to expand in the long axis and the short axis directions, and are penetrated by the arc curve at four corners to form an “O-shaped” closed crack curve; finally, the “X-shaped” cracks are full of the “O-shaped” closed curve. At this time, the four sides of the fixed support plate will break into four geometrically movable rock blocks, the four rock blocks will rotate sink to the goaf like a cantilever beam. With the collapse of the broken roof strata, the broken rock blocks accumulate to be a loose rock body, the loose body then support the sinking roof strata, finally the Voussoir Beam Structure forms, which the Key Block is simplified into an arch with three articulations [1,2,3]. At present, Voussoir Beam hypothesis is widely used in China to design and guide coal mining. According to the hypothesis, the overlying strata movement of stope is divided into three zones in horizontal and vertical direction, as shown in Figure 1. The overburden of stope is divided into caving zone, fracture zone, and bending subsidence zone from bottom to top. In the advancing direction of the working face, Block A represents the Key Block A of the rock mass aggregate and is the mining support stress area; Block B is the Key Block B of rock mass aggregate, which is the area of intense movement of rock strata (fracture development area), and the area of intense change of mining pressure, fracture, and rotate-sinking; Block C is the Key Block C of the rock mass aggregate, which is the re-compaction area. The movement of the rock layer tends to be stable again, and the fracture of the rock layer closes again [2].

However, it still cannot meet the requirements of “white box” to accurately determine the parameters under specific circumstances, which can only be determined qualitatively but not quantitatively [1]. The main reason is that all these theories are based on the hypothesis of mining practice and similar simulation test, the deformation of the surrounding rock of the stope is invisible and imperceptible after the coal is mined out. Therefore, it is a challenge for mining engineers to adopt scientific methods to dynamically monitor the movement of strata and confirm the law of strata movement on the spot. Nowadays the qualitative analysis can only be conducted indirectly by other means through traditional research methods including: empirical methods [5], theoretical calculation [6,7,8,9], physical similar simulation test [7,8,9], numerical simulation [7,10,11,12], field measurement [9,13]. The on-site monitoring of the movement of the strata in the stope is the ultimate, most persuasive means, but it is difficult to monitor the dynamic evolution process of the movement of the roof strata quantitatively by the traditional on-site monitoring methods such as measuring the injected water loss rate along the borehole [14], industrial television endoscope [15], flushing fluid loss method [16], and electric method [17] or combining the above several methods [18] etc., in the field monitoring. What all the traditional field testing methods can do is to monitor the movement results of the rock strata and the final height of the water conducting fracture zone. Actually, these methods improved our understanding degree of the overburden movement, however, they suffer from several substantial defects in practical applications, such as low level of quantification, heavy reliance on personal judgment, and difficulty in reflecting the actual in-situ mining-induced deformation and failure of the overlying rock mass. In particular, these methods do not accurately and dynamically reflect the entire process of strain changes, deformation, and failure of overlying strata over the course of mining.

With the emergence and development of distributed fiber optical sensing (DFOS) technology in the 1980s, optical fiber has been regarded as one of the top inventions of the last decades revolutionizing not only the telecommunications systems, but also new fields of applications such as sensing and metrology [19]. DOFS emerged with the development of a technique called optical time- domain reflectometer (OTDR). There are three main types of OTDR sensing mechanism, which are based on Brillouin [20,21,22,23,24], Rayleigh [25,26,27,28,29], and Raman scattering [30,31].

Compared with the traditional monitoring methods, there are many advantages of DFOS, such as small size and light weight; low transmission loss; security; flexibility; large bandwidth; reliability and low cost; robust; its non-electrical nature making them immune to EM interference and to electrical noise, also allowing them to work into explosive environments; its high sensitivity and wide operating temperature range; especially its distributed measurement; its long-distance and all-round monitoring, so it can be inferred that there are many opportunities and challenges for DFOS in coal mine field. DFOS can be applied to monitor the deformation of overburden mining in the indoor similar simulation test to carry out the dynamic observation of the height of the water conducting fracture zone in the coal mine; to carry out the real-time monitoring of the health status of the underground roadway in the coal mine to prevent the occurrence of roof accidents; to carry out the real-time monitoring of the temperature of the goaf in the coal mine to prevent the occurrence of fire; to carry out the underground mine pressure observation to prevent the rock burst accidents; to monitor the underground water quantity in order to prevent the occurrence of water inrush accidents; to monitor the underground gas content variation by DFOS to prevent the occurrence of gas accidents; and to study the stability of stope and the subsidence of ground surface to protect the ecological environment, etc. Obviously, if the optical fiber network is embedded in the rock strata, just like the perceptual neural network is deployed in the rock strata, the deformation and movement of the rock strata in the “black box” may become a “white box” problem, and various static and dynamic parameters of the rock stratum in mining can be accurately obtained. Therefore, using DFOS technology to monitor the deformation and failure of rock strata affected by mining becomes a research hotspot nowadays. Many use DFOS in indoor similar model physical simulation experiment of coal mining process to study the movement low of overburden of a simulated working face [32,33,34,35,36]. Although many others using DOFS to monitor the movement of overlying strata or overburden caused by underground coal mining. The main result and intention of these tests are mainly the height of water conducting fracture zone caused by overburden movement [37,38,39,40] rather than the dynamic evaluation process of strata movement.

For the sake of solving all kinds of roof problems encountered in the coal mining of Yinying Coal Mine in Shanxi Province, China, a borehole was drilled above the vertical roof strata of the 150,313 working face in the air-inlet roadway, the sensing optical fiber is embedded in the borehole, and the distributed optical fiber strain monitoring technology is used to monitor the movement process of roof strata along with the working face advancing, in order to grasp the movement law of 15 coal’s roof strata, optimize the mining design in time, and ensure the smooth mining of the working face.

## 2. Distributed Fiber Optical Sensor (DFOS) Technology

A distributed optical fiber sensor is defined as an intrinsic sensor that is able to determine the spatial distribution of one or more measurands at each and every point along a sensing fiber. DOFSs use optical fibers both as the sensing element and as the means of carrying the optical signals used for this purpose. [41] DFOS is gaining prominence in the field of structural health monitoring because of several advantages such as high spatial density of data, ease of installation, and reliability [42].

The main stream DFOS widely used in strain monitoring based on Brillouin Scattering technology are the BOTDR, BOTDA, and BOFDA [43]. They have been applied to the strain monitoring of various kinds of structures in many industries, such as high anchored pile wall in gravel [44], Bored Pile [45], Pile Foundation [46], Beam Bridges [47], similar material test model [48], steel rails [49], subsea cables [50], geotechnical structures [51], River Levee Collapse, Concrete Structures and yacht [52], tunnel healthy [53], the state of underground copper mine [54], land subsidence [55], airplane structure health [56], and so on.

Among Brillouin D-FOSs, the BOFDA has the advantages of high spatial resolution (0.2 m), high repeatability and high strain resolution (±2 με), and high measuring precision (±0.002 mm) [57]; while BOTDR is a single ended technique where a light pulse is launched into one end of an optical fiber and the power of the spontaneous Brillouin backscattering is measured from the same end using a spectrum analyzer [58], which has the advantage of long distance (80 km), high resolution (1 m) [58], which is suitable for field monitoring work. Therefore, this paper adopts BOFDA technology in indoor experiments and BOTDR technology in underground field experiments.

### 2.1. The Principle of BOTDR and BOFDA

#### 2.1.1. BOTDR

BOTDR monitoring technology is based on the spontaneous Brillouin scattering (SBS) light in fiber, which can measure the change of temperature and strain along the whole distributed sensing [59]. BOTDR system relies on the fact that the power spectrum of the Brillouin scattered light produced in the optical fiber, undergoes a Brillouin frequency shift (BFS) in proportion to the strain at the scattering position of the fiber [60]. The power spectrum has a Lorentzian shape. Brillouin scattered light is influenced not only by the strain but also by temperature; that is, it also undergoes a frequency shift in proportion to the temperature variation as shown in Equation (1) and Figure 2.
(1)$$v_{B}\left(T,\;\varepsilon \right)\;=\nu_{B}\left(0\right)+C_{T}\left(T-T_{0}\right)+C_{\varepsilon }\left(\varepsilon -\varepsilon_{0}\right)$$
where *T_0_* and *ε*_0_ is the initial temperature and strain, respectively, ν_B_(0) is the initial BFS, *C_T_* and *C_ε_* are the temperature and strain coefficient of BFS, respectively (approximately 1.1 MHz/°C and 0.05 MHz/µε [40]). With the BOTDR, a pulsed light is launched into one end of a single-mode optical fiber, and the Brillouin backscattered light generated by it is observed using a heterodyne receiver at the same end. The scattering position along the fiber is determined by both the light velocity and the elapsed time between launching and receiving the light, that is, in the time domain by using the OTDR, as shown in Equation (2):(2)$$Z=\frac{c\cdot T}{2n}$$

Here *c* is the light velocity in a vacuum, *n* is the refractive index of the optical fiber and the time interval T between launching the pulsed light and receiving the scattered light at the end of the fiber.

In this way, we can obtain the strain distribution along the optical fiber from the observed spectra by using Equations (1) and (2). In Figure 2, we see the correspondence between the strains *ε* and the Brillouin frequency shifts *ν_B_*(ε), respectively. In actual measurements, the coefficient *C_T_*, C_ε_, and *ν_B_*(0) are measured beforehand, and the frequency shift is converted to strain.

#### 2.1.2. BOFDA

The Brillouin optical frequency-domain analysis BOFDA is based on the measurement of a complex transfer function H that relates the amplitudes and phases of counter propagating pump and probe light waves along a sensor fiber [61] as shown in Figure 3.

In BOFDA, the continuous wave light of a narrow-linewidth pump laser is coupled into the left end of a single-mode fiber. At the other end the continuous wave of a narrow-linewidth probe laser is coupled in, whose frequency is shifted compared with that of the pump laser by an amount that equals the temperature and strain-dependent characteristic Brillouin frequency *f*_B_ of the fiber. In this case maximum Brillouin interaction between both waves in the fiber occurs. If the frequency difference between both lasers equals to characteristic Brillouin frequency, the pump light interacts with the modulated probe light in the fiber [62]. Here the probe laser is sinusoidal modulated in amplitude by an electro-optic modulator with a variable modulation frequency *f*_m_. The modulated probe intensity is the boundary condition for the Stokes wave in the fiber. The pump and Stokes waves interact in the fiber by stimulated Brillouin scattering. Therefore, the constant pump intensity at the left end of the fiber gets a sinusoidal changing component. For each value *f*_m_ of the modulated probe intensity, that equals the Stokes intensity Is (L, t) at z = L and the alternating part of transmitted modulated pump intensity Ip (L, t) are detected by photodetectors, whose output signals are fed to an network analyzer (NWA). The NWA determines the amplitudes and phases of both signals, which are digitized and baseband transfer function HL (ω_m_, *f*_D_) of the sensor fiber is calculated, where *f*_D_ = *f*_p_ − *f*_s_ is the frequency difference between the two lasers. The output of the NWA is digitized by an analog-to-digital converter (A/D) and fed to a signal processor that calculates the inverse fast Fourier transform (IFFT). For a linear system, this IFFT is a good approximation of the pulse response of the sensor fiber and resembles the temperature and strain distribution along the fiber. Relationships between temperature, strain, and Brillouin optical frequency drift were used to obtain the information of each measurement point on the fiber, as follows Equation (3) [62]:(3)$$f_{B}=f_{B}(0)+C_{\varepsilon }\cdot \varepsilon +(T-23{}^{\circ}C)\cdot C_{T}$$
where *f*_B_ (0) is a characteristic Brillouin frequency of undisturbed fiber, Hz; *ε* is the relative strain, %; *T* is temperature, °C; *C*ε = 500 MHz/% is strain coefficient, *C_T_* = 1.2 MHz/°C is temperature coefficient.

When the temperature effect is ignored, Equation (3) is simplified to [63]:(4)$$f_{B}\left(\varepsilon \right)=f_{B}(0)+C\varepsilon \cdot \varepsilon =f_{B}(0)+\frac{df_{B}(\varepsilon )}{d\varepsilon }\cdot \varepsilon$$

Therefore the following Equation can be derived from Equation (4):(5)$$\left\{ \begin{array}{l} \frac{\varepsilon }{d\varepsilon }=\frac{f_{B}(\varepsilon )-f_{B}(0)}{df_{B}(\varepsilon )}\\ \varepsilon =-\frac{\Delta f_{B}}{\bar{f_{B}}\cdot Ks} \end{array}\right.$$
where $\bar{f_{B}}$ is the average frequency, *K*s = 0.78. The expression of strain ε can be obtained from Equation (5) [63].

### 2.2. Principle of Optical Fiber Strain Monitoring

From the above analysis of strain monitoring principle of BOTDR and BOFDA, there is a good linear relationship between the axial strain and temperature of the fiber and the corresponding frequency drift for all the BOTDR and BOFDA. So, a simplified general equation can be summed up to obtain the temperature and strain distribution of the whole fiber by calculating the frequency shift of the back Brillouin scattering light, as shown in Equation (6).
(6)$$\Delta \nu_{B}(\varepsilon ,T)=C_{\varepsilon }\cdot \Delta \varepsilon +C_{T}\cdot \Delta T$$
where Δ*v_B_* is the change in Brillouin frequency due to a change in strain, Δ*ε*, and in temperature, Δ*T*. *C_ε_* and *C_T_* are referred to as the strain and the temperature coefficient of the Brillouin frequency shift, respectively.

Another outstanding advantage of Brillouin scattering light compared with other scattering light is that its frequency shift variation and temperature dependence are much smaller than that of strain (20 µε/°C). As the temperature has much less effect on the Brillouin frequency shift than the strain, the temperature change can be ignored or be discounted based on the temperature compensation [64]. A temperature sensor or a free optical fiber free from external force can be used for temperature compensation [65]. Thus, in the indoor experiments, the time period is very short and the temperature is basically constant, the influence of temperature on frequency drift can be ignored; in the underground field test, the free section optical fiber which is not affected by external force served as the communication function is used to conduct temperature compensation for strain measurement. According to the relationship between the strain of optical fiber and the Brillouin frequency shift, the strain of optical fiber can be obtained according to Equation (7)
(7)$$\varepsilon =\frac{\Delta \nu_{B}f_{B}}{\nu_{B}\left(0\right)\times c_{\varepsilon }}=\frac{\nu_{B}\left(\varepsilon \right)-\nu_{B}\left(0\right)}{\nu_{B}\left(0\right)\times c_{\varepsilon }},$$
where *υ_B_*(*ε*) is the Brillouin frequency shift with a strain *ε*, *υ_B_*(0) the Brillouin frequency shift without a strain, *Cε* is *the* coefficient of strain, and *ε* is the strain.

## 3. Indoor Experiments of Fiber Strain Based on DFOS

### 3.1. Experimental on Strain Transfer Performance of SS Fiber in Borehole Anchorage

The damage detection properties of DFOS technique can be defined by evaluating the changes in strain profiles due to a crack formation. These changes are dependent on the cables strain transfer mechanism and the interrogator properties [66]. So, the strain transfer mechanism should be studied before any formal field experiment. In the field test of this paper, the optical fiber is implanted into the borehole drilled through the upper rock-strata, in which concrete grouted and the optical fiber is contained in the anchorage formed after the concrete solidifies. In general, the deformation and failure of rock strata cause the deformation and fracture of anchorage, which are then transferred to optical fiber to generate strain changes. The strain of optical fiber can reflect the deformation and failure of rock strata. In order to verify the above viewpoint and master the strain transfer mechanism of the fiber when the anchor solid breaks, this test is specially carried out.

#### 3.1.1. Experiment Method and Experiment System Setup

First, according to the proportion of concrete pouring in Yinying mine underground, a concrete column of Φ 60 mm embedded with optical fiber in advance is made to simulate the concrete anchorage in the field. Its length is 2 m, and the center of the column is a Φ 12 mm left-handed threaded steel bar, and 5 mm steel strand optical fiber (SS fiber), and 2 m fixed-point optical fiber (FP fiber) were laid out horizontally at a distance along the two sides of the steel bar respectively beforehand. The experiment system setup is shown in Figure 4.

During the experiment, the concrete column is placed on the test platform, and the rebar in the concrete is fixed by two fixed supports on the experiment platform, strain gauges are stocked on the concrete column surface as shown in Figure 4a.

In the pull-out experiment, manual hydraulic pump is used to pull out the reinforcement in the concrete anchorage body, thus causing the concrete structure to deform, as shown in Figure 4b.

The test system of the experiment is divided into three parts: one is BOFDA testing the strain change of the embedded-in optical fibers; the other is testing the strain change of the strain gauges during the concrete drawing process; the third is monitoring the displacement and tension change of the steel bar drawn during the drawing process, as shown in Figure 5.

BOFDA strain tester is produced by Germany fibrisTerre GmbH. Company, the instrument model is fTP2505, its parameters, indices, and working principle are as per [67]. The test in this paper uses a sampling resolution of 0.05 m, a spatial resolution of 0.2 m, a pulse width of 10 ns, a strain test accuracy of 2 με, the refractive index of 1.468, the start frequency and end frequency of the instrument are 10.5~11.0 GHz (this range theoretically, a large strain of 12,500 με can be measured without overflowing the Brillouin frequency), the center frequency of the optical fiber 10.8390 GHz, and a frequency interval of 5 MHz.

In the process of pull-out experiment, the piston of the hydraulic cylinder pulls the reinforcement through the continuous loading of the manual hydraulic pump. The other end of the reinforcement is tightly fixed on the supports on both sides by the anchor lock. The steel bar drives the concrete column to be stretched continuously in the direction of tension.

The method of displacement control is adopted in the test, that is to control the displacement step distance of the steel bar to be stretched well, and the strain data of BOFDA is tested after the strain gauge data is stable after each pull of 1 mm. In the test, there were 22 times of pulling. During the test, the stress in the middle part of the concrete column is concentrated, and cracks appear in the middle part of the drawing process from the beginning to the development of cracks. The cracks gradually increase, and finally the concrete column is broken in the middle part, as shown in Figure 6.

#### 3.1.2. Analysis of Experiment Results

The strain curve of FP fiber measured by BOFDA shows a trapezoid convex platform curve, and the height of the convex platform gradually increases with the axial tension of the concrete. When the axial tension of the concrete exceeds 16.2 mm, the strain step suddenly rises greatly, from 1000 με to 3000 με, and the gradient of the step is 2000 με, which rises to 6000 με when the tension is 22.1 mm, and then stretches with no strain phenomenon, as shown in Figure 7. It shows that the FP fiber and concrete are in coupling state before stretching to 16.162 mm, and the optical fiber deforms with the deformation of concrete; after that, the two are in separation state, and the strain of optical fiber is not controlled by the deformation of concrete until it is pulled off after stretching to 22.1 mm.

Figure 7 shows that the strain change of FP fiber is not consistent with that of concrete column, and there is no peak curve reflecting the stress concentration in the middle part of the column and the damage process. The strain change of FP fiber cannot accurately reflect the failure process of the cylinder drawing process, and also cannot accurately reflect the movement of the rock when it is placed in the drilling anchor solid.

The strain of the SS optical fiber in the concrete is in the shape of an arc boss, and the strain curve is consistent before the whole steel strand is pulled off, which shows that it has good coupling with the concrete, as shown in Figure 8. Before the concrete is stretched 16.162 mm, the middle boss of the strain curve presents regular pattern and gradually rises, with great changes in the middle; later, the middle part of the strain curve rises sharply, and the shape of the strain curve conforms to the actual situation of failure, that is, the middle part of the anchor solid column cracks first, and the middle part of the whole is damaged greatly, which means that both the fiber and the anchorage are at least in the coupling semi coupling state until the concrete is pulled to be broken. When the extension is 30 mm, the strain of optical fiber is still at the level of 2000 με; it is found in the curve that the strain curve at the right end of the curve rapidly reduces to compressive strain, which is due to the concrete fracture at the right drawing end, as shown in Figure 6h. The fracture block compresses the optical fiber, resulting in the increase of optical loss of SS fiber and the disappearance of signal.

Figure 8 shows the process from small crack to large crack in the middle part of the concrete column, and finally the middle part fractured and broken at last. That is to say, the peak value of optical fiber in the middle part of the column decreases gradually to both two column ends; with the column being pulled the damage in the middle part is intensified, and the peak value curve is increasing. It can be concluded that the SS fiber can accurately reflect the failure and deformation of concrete anchor in the borehole, and then can accurately characterize the movement of the rock strata when it is placed in the drilling anchor solid.

This also can be seen through the Figure 9, it is obvious that the strain curve of SS fiber and strain gauge is in the same order of magnitude, with high coupling performance; after drawing 16 mm, the strain gauge curve of FP fiber shows a straight upward trend, which is different from that of strain gauge, which also shows the bad coupling of deformation of FP fiber and concrete in the meantime. The experiment shows that the SS fiber with higher strength has higher tensile and breaking resistance, which is suitable for implanting in the rock strata to characterize the deformation of the rock strata.

From Figure 7, Figure 8 and Figure 9, the three strain changes of the fiber in the concrete anchorage can be used to accurately represent the movement of the rock strata. ① When the fiber is in the initial strain state and the strain peak change is small, although the anchor solid may have slight strain change under the action of tension, the damage of the anchor solid is small, and the surrounding rock stratum where the anchor solid is located is in the original rock movement state or in a state of strata movement as a whole. ② When the peak strain curve appears in the optical fiber, the anchor solid failure at the peak position indicates that the position is in the separation position, and the strata of the position has been broken, strata on two sides of the peak strain belong to different rock block; or it indicates that the lower rock layer collapses; or it indicates the horizontal shear movement of the upper and lower rock strata. In a word, the peak strain of optical fiber indicates the vertical separation of the rock or the horizontal displacement of the rock causes the horizontal shear failure of the anchor.

### 3.2. Strain Transfer Performance Test of SS Optical Fiber

#### 3.2.1. The Introduction of the SS Fiber and FP Fiber

According to the above tests, the distributed optical fiber monitoring of roof strata mining deformation mainly uses SS fiber. At the same time, FP fiber is used for reference and communication.

SS fiber is the common name of Metal Based Cable (nzs-dss-c02) produced by Nanzhi Company. The SS fiber is composed of four layers as shown in Figure 10a. From the inside to the outside, it is composed of bare fiber, PVC sheath, metal reinforcement tightly wound around the center and high-strength plastic protective layer. The sensing optical fiber with sheath is distributed in the center in a straight line, and six metal wires (ribs) are tightly wound in a spiral shape around it. This kind of structure can effectively enhance the tensile strength, bending resistance, and friction impact resistance of the SS optical fiber, and greatly improve the survival rate of the layout of the sensing optical cable.

The so-called FP fiber, that is, by artificially applying fixed points on the sensing fiber, and the deformation between two adjacent fixed points can be sensed by the fiber. The strain of the sensing fiber between the two fixed points is not affected by the deformation of the surrounding environment of the fiber, but depends on the relative displacement between the two fixed points. When the FP fiber is laid, it is usually just to stick or fix the aluminum alloy tubes on the surface of the structure under test shown as Figure 10b. This kind of fiber can solve the problems of fixed point method in monitoring large deformation of rock mass.

The specifications and parameters of SS fiber and FP fiber are shown in Table 1, and their physical pictures and structures are shown in Figure 10.

The SS fiber uses high-strength steel wire to protect the delicate and fragile sensing optical fiber, forming a tight sleeve distributed sensing optical cable with convenient installation, high tensile strength, and strong bending resistance.

In order to accurately reflect the strain and deformation of roof strata, it is necessary to calibrate the corresponding sensing coefficient of the SS fiber before the actual test in the field, then to study the strain transfer performance of SS fiber, finally to determine whether necessary calibration is carried out.

#### 3.2.2. Calibration of SS Fiber Strain Coefficient

The test was carried out with a tubular optical fiber stretching device made by Suzhou Nanzhi Company. Two ends of the stretching device are fixed with special clamps, one end is fixed on the outer wall of the device, and the other end is fixed on the outer edge of the moving valve of the hydraulic pump, as shown in Figure 11. The manual hydraulic pump drives its hydraulic valve to move to pull the optical fiber, which makes the SS fiber produce strain. The Brillouin frequency shift (BFS) data of SS fiber is measured by BOFDA with high accuracy, and the measured data of SS fiber strain is obtained by electronic dial indicator fixed at both ends of the stretching device.

The test adopts the method of drawing 2 mm each time, and the test data are shown in Figure 12.

The above test process is repeated for 5 times. The BFS of SS fiber under different stretching distance is taken as the average value, corresponding to the strain of optical fiber caused by different stretching distance, and the strain BFS curve is obtained, as shown in Figure 12. The strain coefficient is 0.04954 MHz/με by linear fitting method, and the standard deviation is 0.99824, which is very close to 1 which shown as Figure 13. The fitting results meet the requirements, close to the theoretical value of the strain coefficient of 0.05 (that of a standard fiber).

#### 3.2.3. Experimental on Strain Transfer Performance of SS Fiber

In order to determine the strain transfer efficiency from the structure under testing to SS fiber, the strain transfer performance test of SS optical fiber was carried out. A bare fiber was pasted on the outer edge of SS fiber sheath, the strain of bare fiber varies when the SS fiber is stretched, the strain difference or strain transfer efficiency of SS optical fiber was tested when SS fiber was stretched to produce strain. Because in the real field strain test of the structure, the SS fiber is implanted or adhered to the surface of other structures, the strain test is the deformation of the structure, which is transferred to the optical fiber sheath to produce strain, and then the deformation of the sheath is transferred to the optical fiber in the inner core to produce strain. Therefore, the consistency of the sheath strain and the core fiber strain is the basic condition of strain test. So, in this experiment, one is to verify the consistency of the strain between the fiber sheath and the core fiber; the other is to show the consistency of the deformation of the structure and the deformation of the fiber when the coupling between the fiber and the structure is consistent. In the test, the above-mentioned optical fiber strain calibration test device is used, it can be refereed to Section 3.2.2. Two kinds of optical fibers were cut with a certain length, first the outer surface of SS optical fiber is cleaned, and a bare fiber was stick onto its surface with epoxy resin, as shown in Figure 14.

When the test fibers’ combinations are ready, they are fused in a series. Other test conditions are the same as strain calibration experiment. In this strain transfer test, the tension step distance of the SS optical fiber is set as 1 mm. After each stretch of the SS fiber, it stands still for 10 min. After the tension is stable, the corresponding strain of two kinds of optical fibers BOFDA and the stretch amount of SS fiber with dial indicator were measured. The test results are shown in Figure 15.

The strain transfer coefficient is defined in two forms.

Form 1 is Equation (8), Form 2 is Equation (9).
(8)$$\zeta =\frac{\varepsilon_{a}}{\varepsilon_{b}}$$
(9)$$\zeta =1-\left|\frac{\varepsilon_{a}-\varepsilon_{b}}{\varepsilon_{b}}\right|$$
where, ξ is the strain transfer rate of optical fiber, *ε*_a_, the strain of SS fiber. *ε*_b_, the strain of bare fiber.

It can be seen from the definition that the closer the defined value is to 1, the better the strain transfer rate is. According to the definition and test, the curve of strain transfer rate is obtained. The test results show two kinds of strain transfer coefficient curves, as shown in Figure 16.

Through the analysis of the test data, the average values of the two definitions are 0.9968 and 0.9937, respectively, which shows that the strain transfer efficiency of SS fiber is quite high, and the interference influence caused by the fiber sheath is very small, and can be neglected. As can be seen in Figure 16, with SS fiber being stretched, the strain transfer coefficient begins to decrease gradually. When SS fiber is stretched to 2 mm, it rises gradually, and the strain transfer coefficient is stable above 0.99, which shows that this is because SS fiber has a pre-stretching effect. When the fiber is stretched to 2 mm, the effect of pre-stretching on strain transfer is weakened or eliminated. According to the overall pull theory, the strain changes of the two kinds of fiber are basically consistent, also can be seen as shown in Figure 17. It can be concluded that when the coupling of the SS fiber and the measured structure is consistent, the strain of the fiber can accurately reflect the deformation of the structure. The strain transfer efficiency of SS fiber meets the test requirements.

As in the previous section, we have verified that the coupling process of SS optical fiber and concrete anchor solid is consistent through the coupling test of SS optical fiber and concrete anchor solid, and Figure 9 also shows that the strain changes of SS optical fiber and strain gauge are consistent. The point of view of this paper is that the concrete anchor solid is drilled into the rock strata in the field, and the failure deformation of the anchor solid reflects the failure process of the rock strata. That is to say, the SS optical fiber deployed in the anchor solid can reflect the deformation and failure of the rock strata. It can be divided into two aspects, (1) when the strain changes little in the strain section of the optical fiber, it indicates that the rock stratum where the optical fiber is located is in the overall deformation state due to the external force; (2) when the optical fiber has a large peak strain curve, the rock stratum at the peak position of the optical fiber is in the state of separation or horizontal shear failure, and the rock stratum at both sides of the peak respectively belongs to a different whole rock block in different movement states.

### 3.3. Introduction to the Instruments Selected for Field Experiment

The BOTDR analysis instrument of this test is av6419 produced by China scientific instrument company. Its parameters are shown in Table 2.

## 4. Distributed Optical Fiber Monitoring of Roof Strata Deformation

### 4.1. The Working Face and Roof Conditions of Field Experiment

Yinying Coal Mine is located in the northeast of Qinshui Coal field, Shanxi Province, China, mainly mining No. 15 coal. The 150,313 working face adopts the comprehensive mechanized caving technology, and the total caving method is used to manage the roof. The strike of the working face is 980 m long and the inclined length is 228 m. There is one air inlet and one air return roadway respectively. The east side of the air return roadway is the goaf of the previous working face, and the west side of the air inlet roadway is the solid coal of the next working face that is not mined. The lithology of the top and bottom slabs and the fiber layout are shown in Figure 18.

The direct roof of 15 coal seam is ① K2 limestone composite roof with large thickness, which is composed of three layers; the basic roof is ② ③ shale and siltstone composite roof; the overlying ④ medium-grained sandstone layer and ⑤ fine sandstone layer are the first composite sub Key Layer of 15 coal seam roof.

### 4.2. Layout of Optical Fiber and Construction of Monitoring System in Field Test

The optical fiber in the roof strata is implanted by drilling a borehole and grouting in the borehole. The specific steps are as follows:A 70-m long drill hole of Φ 60 mm was set at the top angle of the coal wall near the working face between the 698 rows and 697 rows’ W steel strip of the air intake roadway of the 150,313 working face. As shown in Figure 19a, the elevation angle of the drill hole is 45°, which is perpendicular to the advancing direction of the working face.Binding SS and FP fibers on the outer wall of Φ 40 mm PVC pipe to keep the optical fiber straight. At the first end of PVC pipe, SS fiber and FP fiber are welded together. The welding position is protected by a hard conical shell, and FP optical fiber is bound on the other side of PVC pipe, as shown in Figure 19b.After cleaning the inner wall of the borehole with drill pipe, PVC pipe bound with two kinds of optical fibers was inserted into the borehole, as shown in Figure 19c.After all PVC pipes with fiber are put into the drilling hole, a special stopper is used to block the drilling hole to prevent PVC pipes from sliding out. The concrete slurry is pumped into the borehole until the slurry flows out of the PVC pipe, as shown in Figure 19d.The optical fiber path was examined throughout the construction. After grouting and sealing, the sensing optical fiber was properly placed in a safe position. It is fused to the communication optical fiber in the roadway, and then the communication optical fiber is arranged to the monitoring location; power distribution center in the main roadway. The configuration of the distributed optical fiber roof deformation monitoring system of the overall BOTDR is shown in Figure 20.

### 4.3. Field Data Collection

Four months after the concrete grouting and hole sealing, the 150,313 working face was mined. At this time, it was 300 m away from the borehole, the initial measurement of BOTDR strain was carried out. When the working face is 100 m ahead of the borehole, the strain data collection begins. When the working face is close to the monitoring position, the data are collected intensively with the frequency of 1–2 times a day. A total of 40 groups of field data were collected, and the monitoring range was −80~10 m.

## 5. Analysis of Sensing Optical Fiber Strain Monitoring Results

The sampling parameters used in the actual test of downhole optical fiber strain are shown in Table 3 below.

### 5.1. Spatial Position Calibration of Sensing Optical Fiber in Coal Mines Underground

When the sensing optical fibers are arranged in the roadways and boreholes, different kinds of optical fibers, under different stress, also present different strain states, which can roughly distinguish the one-dimensional position of the sensing optical fiber strain. However, it is necessary to accurately determine the spatial position of the strain in the sensing fiber. The three-dimensional spatial position calibration of optical fiber becomes an important monitoring procedure step for accurately positioning the different rock strata with different lithology.

#### 5.1.1. Spatial Position Calibration Using Initial Strain in Field Test

As described in Section 4.2 step 1, the SS fiber and FP fiber are fused to be a whole fiber as a loop in the borehole, and is defined as test line ① from the SS fiber to the FP optical fiber, and test line ② from the FP optical fiber to the SS optical fiber. Using the loop method, even if the SS optical fiber breaks in the borehole, we can still obtain the strain curve of SS optical fiber from two directions to deduce the strain curve of the whole SS optical fiber in the borehole, and then study the movement of roof strata. According to the strain theory discussed above in Section 2.1, the drilling position can be roughly framed by the original strain curve, as shown in Figure 21.

The original initial strain curve from the SS fiber to the FP fiber direction is shown in Figure 21. It can be seen in the figure that 0~884 m is communication common optical fiber, 884~1257 m is 5 mm SS optical fiber, 1257~1332 m is FP optical fiber in the borehole, so it is impossible to accurately determine the position of SS optical fiber in the borehole; moreover, the positioning length of FP optical cable reaches 75 m, and the error between the actual borehole length of 70 m is as much as 5 m. The error caused by this method of location based on the original strain curve makes the strain curve unable to accurately reflect the deformation movement of special roof strata in different lithology strata, let alone quantitative analysis. In order to achieve the accurate characterization of the law of movement of the strata with optical fiber strain, the accurate position calibration of the sensing optical fiber is the primary, most important, and decisive work of roof strata movement monitoring.

The underground environment of coal mine belongs to the explosion-proof environment. The conventional chemical condensing agent quick freezing method, electric heating method, and so on all violate the rules of coal mine production, so it is necessary to explore a new method for calibration.

#### 5.1.2. Indoor Test of Spatial Position Calibration by Hot Stick Method

According to the principle of temperature change and Brillouin frequency shift of optical fiber, in the laboratory of China University of mining and technology Beijing, our research group adopted the heating stick that can keep the temperature of infusion bottle to test the temperature effect of the heating stick on the optical fiber. The heating stick is composed of heating mineral materials. When the protective film is torn off, it will generate heat when encountering air, and will not produce sparks and electric current, etc., which meets the conditions of underground coal mine. It was found that it takes 20 min to reach the maximum temperature of 87 °C after the application of hot compress, the duration of the maximum temperature was 70 min, and the time of keeping above 70 °C was 100 min. When the fiber length is more than 1 m, and it is fully pasted with the hot sticks, it is found that the fiber strain rises 900 με using BOTDR monitoring the strain changes. The test process and test results are shown in Figure 22.

It is found that the strain change is not obvious when the length of hot deposit is less than 1 m. When the hot deposit length is more than 1 m, the error between the midpoint position (peak value) of the strain curve and the midpoint position of the hot deposit length is less than 10%, i.e., less than 10 cm, which meets the requirements of the coal mine underground field test. In the test, not only the high temperature is more than 80 °C, but also the temperature is reduced to 40 °C for more than 2 h, the BOTDR strain can still accurately reflect the hot deposit position.

#### 5.1.3. Field Experiment of Optical Fiber Spatial Position Calibration

In order to achieve the accurate representation of the law of deformation and failure of the strata with optical fiber strain, the accurate calibration of the three-dimensional space position of the sensing optical fiber is the primary work of roof strata movement monitoring. As shown in the Figure 18 above, since the optical fiber in the borehole is a loop, the test line ① is defined from the SS optical fiber to FP optical fiber direction, and the test line ② is defined from the FP optical fiber to the SS optical fiber direction. Therefore, a 1.5 m long hot stick pack was applied on the SS optical fiber at the drilling orifice location, the distance between the middle point of the hot deposit length and the calibrated position of the orifice is 4 m, as shown in Figure 23.

After half an hour of hot laying, strain test was conducted for the whole monitoring optical fiber line through BOTDR. Line ① and line ② are tested twice respectively. The strain curve of optical fiber is shown in Figure 24.

From the peak point of the strain curve, the middle position of the hot deposition length is obtained. The initial orifice position of the line ① is 1181.32 m of the optical fiber length; the initial orifice position of the SS fiber of line ② is 1346.9 m of the optical fiber length. According to the length of the borehole and the length of the optical fiber fed into the borehole, the exact position of the optical fiber at the end of the borehole and the corresponding position of the optical fiber and different strata can be obtained.

### 5.2. Strain Distribution and Variation Characteristics of Optical Fiber in the Borehole

The monitoring hole is located at the coal wall side of the roadway, perpendicular to the advancing direction of the working face, with an elevation of 45° and a drilling length of 70 m. The strain of the fibers mainly represent the deformation characteristics of the overburden within 49.49 m of the vertical direction of the roof, including the direct roof, the basic roof, and the nearest Key layer, as shown in Figure 14. The optical fiber test of the borehole is controlled by an optical switch.

#### 5.2.1. The Fiber Strain Curve in the Direction of Line ① in the Borehole

Figure 25 shows the strain distribution curve of test line ①. Its monitoring sequence is from the SS fiber at the initial point of the orifice to the bottom of the hole, from the bottom of the hole to the FP fiber, and then from the FP fiber to the orifice, connecting communication fiber at last.

Within the hole depth of 10 m, it can be seen in Figure 20 that when the working face is close to the hole opening, the optical fiber at the position of top coal, pseudo roof, and direct roof fluctuates between −800~+500 με under the comprehensive influence of leading support deflection, top coal caving, and tunnel convergence, with complex changes. The length of strain curve of FP fiber is basically the same as that of SS fiber, but the strain curve of FP fiber mostly presents irregular zigzag curve. The fiber strain of this part of length has little to do with the analysis of roof strata movement law, which is not the focus of this paper. So, this paper mainly discusses the strain change of SS fiber at the orifice height of more than 10 m to reveal the movement of roof strata. The process of optical fiber monitoring and the change of optical fiber strain curve are as follows:

The overall strain curve is close to the initial strain state before the working face passes through the borehole.

On 7 September, the working face passed the borehole, two small double peaks of 200 με appeared at the position 30~40 m away from the orifice depth of the hole.

On 8 September, after the working face passed the hole for 4 m, the initial strain state of the SS fiber was maintained at the orifice height from 10 m to 25 m, and a multi-stage boss curve with a peak value of 2000 με appeared from 25 m to 51 m, and the peak value was located at 35 m of orifice; the strain curve of the SS fiber was maintained at the platform curve section with a peak value of 1000 με between 40~51 m of the orifice height. From the 51 m depth of the hole, the strain rapidly decreased to the initial strain state when the orifice height was 63 m, and reached the bottom of the hole.

On 9 September, the strain curve was basically the same as that on 8 September. After the working surface passed the test hole for 7 m, except that the maximum peak tensile strain of SS fiber at the hole depth of 35 m reached 3240 με, the strain platform at the first level of 2000 με was at the hole depth of 40 m~51 m, and the strain platform at the second level of 1000 με was at the hole depth of 53~63 m. From the hole depth of 63 m, the strain of the optical fiber gradually decreased to the initial strain at the hole bottom.

In the morning of September 10, when the working face pushed through the borehole for 9 m, the strain of SS fiber suddenly increased at the hole depth of 12 m, and at the hole depth of 18 m, the strain of the optical fiber reached a 3000 με tensile strain platform, with the peak value at the hole depth of 35 m, reaching 4173 με, and then when the platform was maintained to the hole depth of 51 m, the optical fiber strain signal was terminated at 11:00 pm. on 10 September, a −500 με fiber appeared within 5 m of the hole depth. The peak value of strain is then transformed into tensile strain rapidly, reaching 500 με at the hole depth of 8~10 m, and then the strain signal disappears suddenly.

#### 5.2.2. Strain Change in the Direction of Line ② in the Borehole

Figure 26 shows the strain curve from FP cable to SS fiber of line ②. It can be seen that the change of strain before 10 September is the same as that of line ①.

The differences of fiber strain from the experiment of line ① are as follows:In the morning of 10 September, the SS fiber lost the strain signal at 7 m from the orifice, kept a small platform of 2000 με between 5~18 m of the hole depth, and suddenly pulled up to 3000 με platform at 18 m of the hole depth. The peak value appeared at 35 m from the orifice. Other changes were the same as the strain change of line ①, but the strain value of 53–63 m increased very little. In the morning of 10 September, at the hole depth of 5 m, the optical fiber strain suddenly rose to the saw-tooth platform above 1000 με, and then the optical fiber strain rose to the saw-tooth platform of 2000 με above the hole depth of 10 m, and kept to the hole depth of 42 m; there was a saw-tooth curve platform of 1000 με between the hole depth of 42~55 m; there was a saw-tooth curve platform within 15 m from the hole bottom irregular curve platform with a maximum of 500 με.At 11:00 pm. on 10 September, when the working face passes through the borehole for 10 m, the strain of FP fiber at the orifice suddenly rises, and then soars to 5000 με at the borehole depth of 5 m, and then maintains this strain height, and suddenly fell back to the zero strain state at the borehole depth of 8 m, which obviously shows that FP fiber is subject to strong tension compression conversion at the borehole depth of 5~8 m, and finally is also damaged by shear.

## 6. Analysis of Mining Movement Law of Roof Strata

As mentioned above, the coupling between SS fiber, rock strata, and concrete anchorage is good, and the change of strain curve of optical fiber is obvious and its content is rich, which can more accurately reflect the law of rock movement. According to the two survey line directions, the strain curve of SS fiber of survey line ①, the strain curve of SS optical fiber of survey line ② in the opposite direction are corresponding to the histogram of roof strata, and the deformation and movement of strata in different strata with the mining of working face are obtained, as shown in Figure 27.

Under the influence of coal mining, the movement processes of roof deformation and failure are as follows:

Before the working face passes through the drilling on 7 September, the overall strain of the optical fiber in the roof strata presents the initial strain state, which indicates that under the protection of the hard and thick limestone and sandstone roof, the roof strata are not obviously affected by the leading support stress. On 7 September, the working face passed the drilling, the optical fiber in the strata of ② ③ roof showed slight tensile strain, indicating that the direct roof of K2 limestone under the optical fiber position had started to collapse layer by layer, causing the rock layer of optical fiber position to become a cantilever beam state. The cracks in the rock began to develop, and there was a slight separation between the layer and the overlying strata. However, there was no obvious tensile strain change in the optical fiber in ① K2 limestone composite roof strata, which indicates that the roof strata first collapsed in the center of goaf, and under the action of solid coal clamping, the direct roof strata at least within the hole depth of 27 m, that is to say, at this time, the roof strata at the position of optical fiber within 19.1 m distance from the inner working face direction of the air inlet wall remained in stable cantilever state. The upper layer of K2 limestone exceeded the ultimate strength limit and collapsed at least 19.1 m away from the air inlet in the middle of the working face, as shown in Figure 28.

On September 8, there was a peak curve of optical fiber in ② shale and ③ fine silt sandstone, and the peak point of 2000 με was in ② shale, which indicated that the optical fiber in this section of rock with soft lithology was strongly tensioned, the vertical displacement of ② rock was relatively large, and it separated from ③ fine siltstone greatly; at the same time, the optical fiber in ③ fine siltstone formed a 1000 με tensile strain platform. It shows that the vertical fracture of the rock strata forms the masonry beam, and the rock block at the position of optical fiber is in the Key Block B of the rotary sinking masonry beam; the optical fiber strain in the overlying ④ medium-grained sandstone stratum is the curve gradually decreasing to the initial strain state, which shows that it is the sub key layer with relatively hard lithology in the state of cantilever beam with slight deflection bending under the action of gravity and mining stress. The optical fiber in No. ⑤ fine-grained sandstone layer kept the initial strain state, which shows that the influence of mining stress on this layer is very low. The change of rock strata in the section of optical fiber position is shown in Figure 29.

On 9 September, the movement of rock strata reflected by optical fiber strain is divided into five parts after the working face passes through the borehole for 7 m. ① K2 limestone kept its initial state of surrounding rock stability. The rotation subsidence displacement of ② shale increased, and the separation with overburden ③ increased; the peak strain of optical fiber in shale increased from 2000 με to 3000 με. The strain of the optical fiber in the ③ fine silty sand layer increased from 1000 με strain platform curve to 2000 με, which indicates that the rotation subsidence displacement increases under the influence of gravity and mining stress. The optical fiber strain in ④ medium sand stratum also formed a strain curve platform of 1000 με, which indicates that ④ stratum had also vertically broken and formed the key block B of masonry beam structure, and the interior is also developed with cracks. The strain of optical fiber indicates that the strata ⑤ and the overlying strata were in the state of cantilever beam. All are showed in Figure 30.

In the morning of 10 September, the working face passed through the borehole for 9 m, and the strain curve showed that except that the collapse of No. ② shale continued to increase, the strain of optical fiber in ①~③ showed a strain platform curve of 3000 με, indicating that the three layers of strata changed from rotation settlement in the early stage to translation subsidence or reverse subsidence (rock block of vertical arrow in Figure 31); the original unbroken cantilever rock layer broke twice and formed a new one key block B (rock block of horizontal arrow in Figure 31). As a result, the optical fiber in the three layers of strata is stretched by the collapse of the strata, and the strain change increased; the breakpoint of the optical fiber strain curve shows when the strata caved, the optical fiber was squeezed by the horizontal force of the strata in the lower part of the K2 limestone layer and the upper part of the ③ fine siltstone layer, combined with the test results of line ②. At this time, the optical fiber strain change of ④ medium-grained sandstone did not change greatly with the mining of working face, and the deformation of ④ rock layer was very small, which indicates that ④ rock layer was strongly supported by the underlying caving rocks in the early stage, formed a stable masonry beam structure, and the optical fiber part in the rock layer was in the re-compaction area of the goaf, as shown in the red double dotted box in Figure 31. It also shows that the overlying layer ⑤ kept the original rock state.

In the night test on 10 September, the location of the optical fiber break point in line ① was the same as that in line ② in the morning of 10 September, that is to say the lower part of K2 limestone layer, it indicated that the optical fiber at this location was bent too much and light-loss in fiber was too large under the action of gravity and horizontal shear stress of the broken rock blocks.

It can be concluded from the overall change of optical fiber strain and stratigraphic correlation map that when the working face does not pass the borehole, the optical fiber presents the initial strain state and the rock stratum is in the original rock stable state. When the working face passes through the borehole for 2 m, affected by the collapse of the underlying top coal and part of the direct roof, the unbroken roof presents a cantilever beam state under the influence of gravity and mining, fissures develop in ② shale and it separate from the overlying hard rock stratum, resulting in the tensile strain change of the optical fiber. After the working face is drilled for 5 m, the ② ③ rock stratum changes to the masonry beam state, and the rock stratum at the optical fiber position is the key block B, the key block B rotates and sinks as a whole. Because of the different rock properties, stratum ② and ③ are separated from each other, and the position of optical fiber in ② shale rock is the position of rock fracture and subsidence, which results in the peak strain of optical fiber, and forms the strain platform curve of optical fiber in ③ rock stratum, and the cantilever beam state of ④ hard rock stratum slightly separated from the layer. After the working face passes through the borehole for 7 m, the rotation and subsidence of ② ③ stratum are intensified, and the rock block at the optical fiber position of ④ stratum is fractured and transformed into the key block B of masonry beam. After the working face passes through the borehole for 9 m, the key block A (or the original rock cantilever beam rock block without fracture) of ② ③ rock layer masonry beam is broken twice, and the key block B is transformed from rotary sinking to translation or reverse sinking and into the key block C of masonry beam, forming the re-compaction area of goaf. The area affected by the secondary fracture of the optical fiber is large, which causes the strain of the optical fiber in ①~③ strata to rise greatly. Later, under the shear action of the newly formed key block B, the fiber strain signal disappears

## 7. Conclusions

In order to explore and study the law of mining deformation movement of roof strata in 150,313 working face of Yinying Coal Mine, the fiber optic strain field experiment and corresponding indoor experiments were carried out. Through these series of fiber strain experiments and strain results analysis, the following conclusions are obtained:(1)Strain coefficient calibration and strain transfer test show that high strength SS fiber has high strain transfer performance. The fiber strain test of concrete anchorage pull-out shows that the coupling performance of SS fiber with the concrete is high in the stage of tension and fracture. The SS fiber implanted into the rock stratum can reflect the deformation state of rock strata better with the deformation of the concrete anchorage, and its application in the roof strata movement monitoring is feasible.(2)It is the first condition to accurately determine the movement state of roof strata to accurately demarcate the three-dimensional spatial position of optical fiber in coal mine underground. It is feasible to demarcate the spatial position by the method of temperature calibration of infusion bottle hot sticking.(3)The results of optical fiber strain show that with the advance of the working face, the roof strata of the goaf first break down in the middle of the goaf. Then the breakage develops upwards and to the two sides of coal pillars in horizontal direction.(4)In the field experiment, it is can be concluded that there are three movement states of roof strata in gob. First, the initial optical fiber strain state represents the strata is in a stably original state; second, the abrupt strain curve with peak value indicates the strata broke and fractured, it means there is a large separation in the position, or the strata below the fiber collapsed; the third, the platform curve of fiber strain indicate that the strata with the same strain value move overall as a whole.(5)Through the analysis of the strata position, strata lithology and the fiber strain, the difference of movement state of strata can be demonstrated in 150,313 working face, that is to say, there are three movement stages of roof strata movement. (1) When the coal is mined out in the goaf, the roof strata first is in cantilever beam state, it sink under the weight of strata; (2) when the soft underlying rock stratum collapse like shale. The broken shale rock blocks fill the goaf with the advance of working face, the broken blocks accumulate to support and prevent the upper hard sandy rock strata from sinking, as a result, the upper sandy rock strata are in masonry beam structure state. (3) In the end, from the K2 limestone to upper medium-grained sandstone near borehole broke twice, the new masonry beam structure emerges, and the old masonry beam collapsed in translational subsidence state. QIAN’s theories are confirmed and verified in the fiber strain field monitoring test based on BOTDR in the meantime.

## Figures and Tables

**Figure 1 sensors-20-01318-f001:**
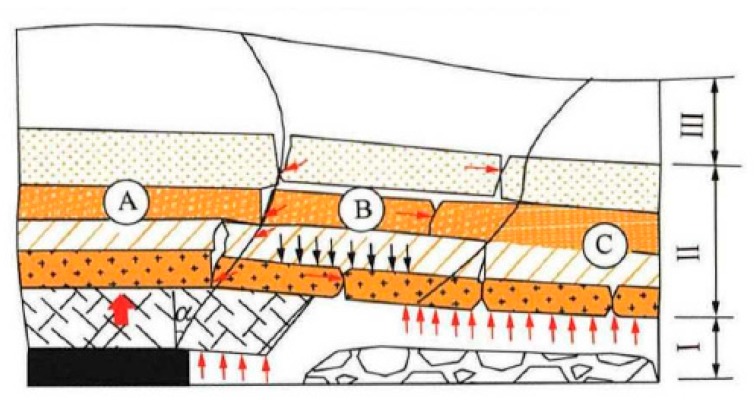
Movement division of overlying strata. I: caving zone; II: fractured zone; III: curved subsidence zone [2].

**Figure 2 sensors-20-01318-f002:**
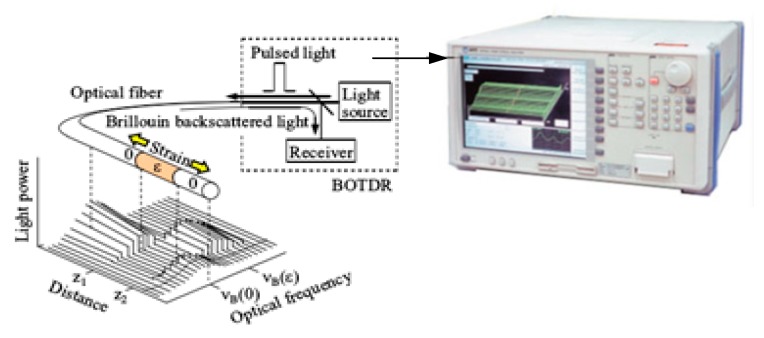
Basic principle of a BOTDR sensor.

**Figure 3 sensors-20-01318-f003:**
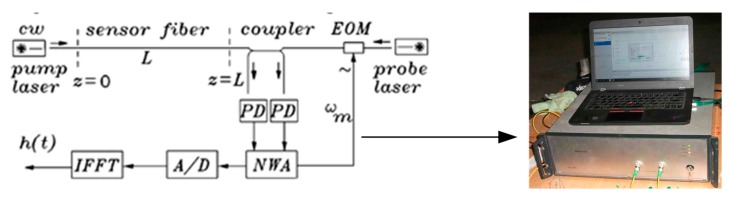
Basic configuration of a BOFDA sensor.

**Figure 4 sensors-20-01318-f004:**
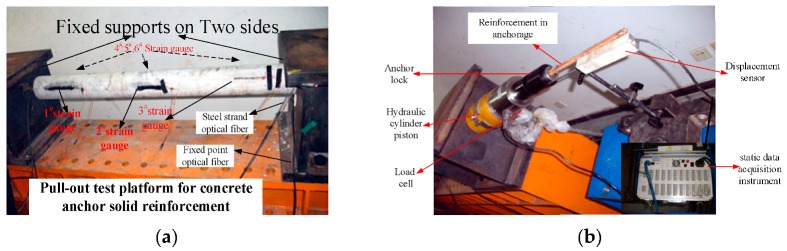
Layout of reinforced concrete pull-out test; (**a**) shows the experiment platform; (**b**) shows the testing system and pull-out system of the experiment.

**Figure 5 sensors-20-01318-f005:**
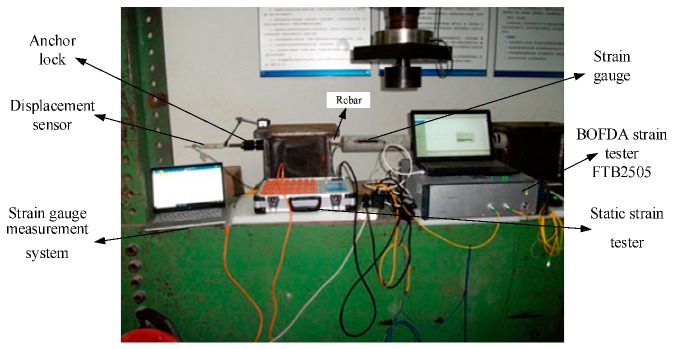
Test system of concrete drawing experiment.

**Figure 6 sensors-20-01318-f006:**
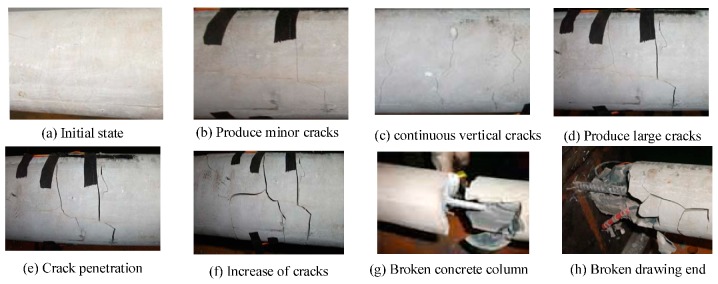
Failure process of reinforced concrete anchorage in pull-out test.

**Figure 7 sensors-20-01318-f007:**
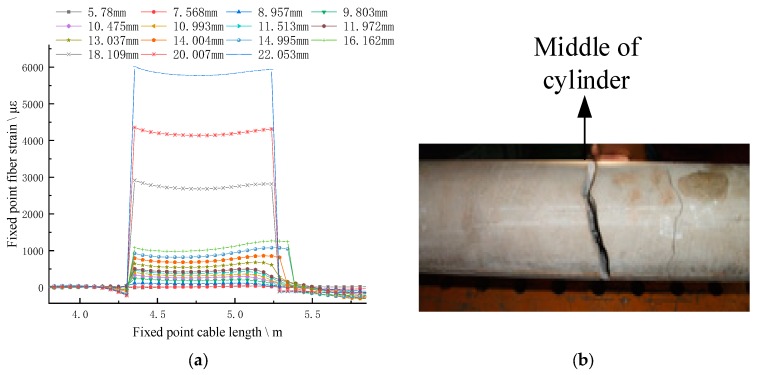
Strain curve of FP fiber (**a**) compared with the actual anchor drawn deformation. (**a**) The strain curve of FP fiber; (**b**) the actual deformation of concrete column.

**Figure 8 sensors-20-01318-f008:**
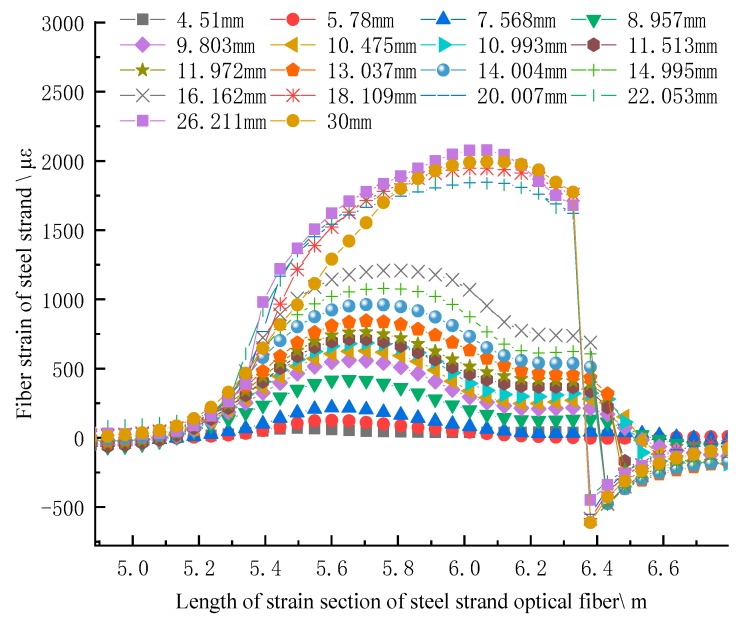
Strain curve of SS fiber under different length of anchor solid drawing deformation.

**Figure 9 sensors-20-01318-f009:**
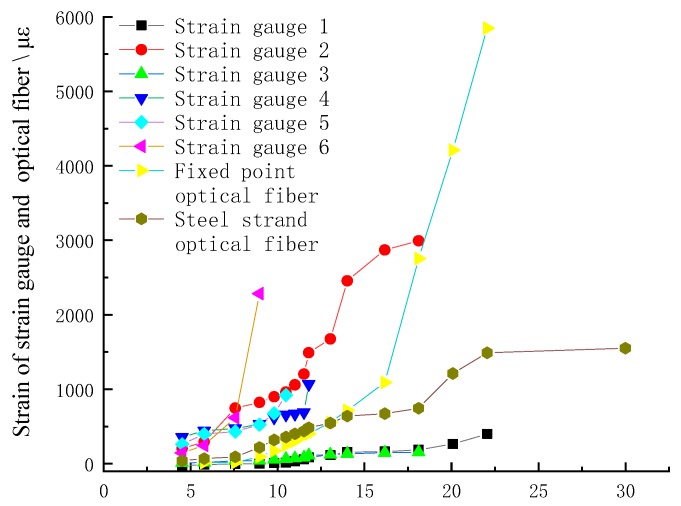
Strain comparison between strain gauge and fiber.

**Figure 10 sensors-20-01318-f010:**
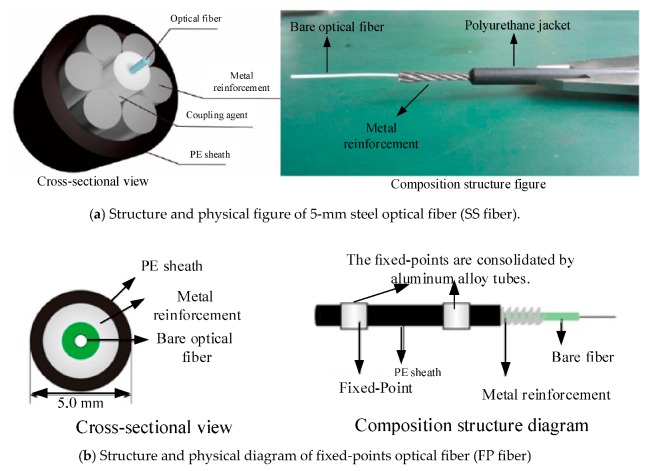
Cross-section and composition structure of the fibers. (**a**) SS fiber and (**b**) FP fiber.

**Figure 11 sensors-20-01318-f011:**
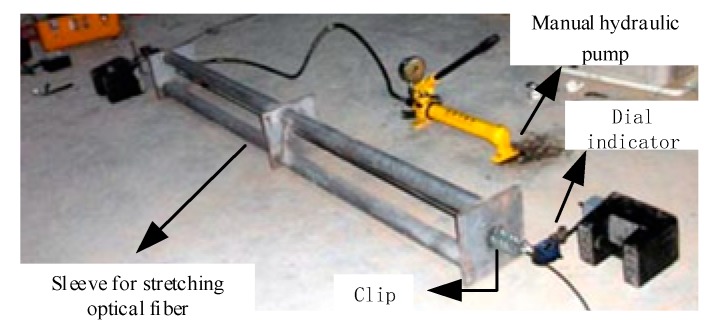
Optical fiber drawing device platform.

**Figure 12 sensors-20-01318-f012:**
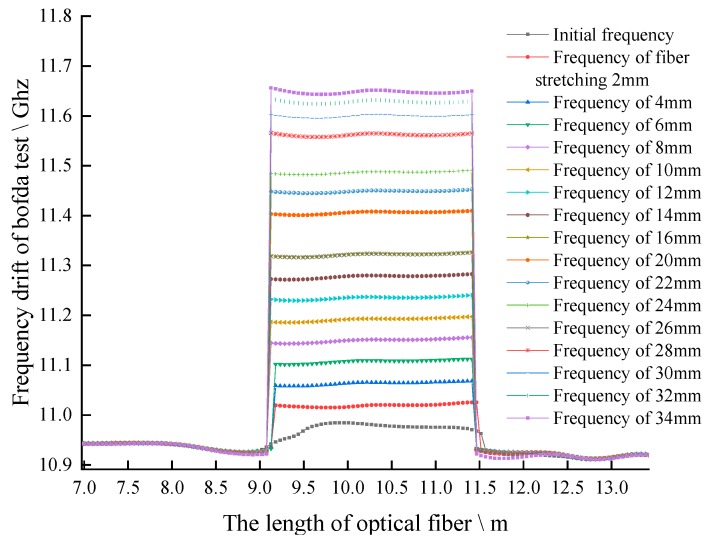
Original Brillouin frequency shift diagram.

**Figure 13 sensors-20-01318-f013:**
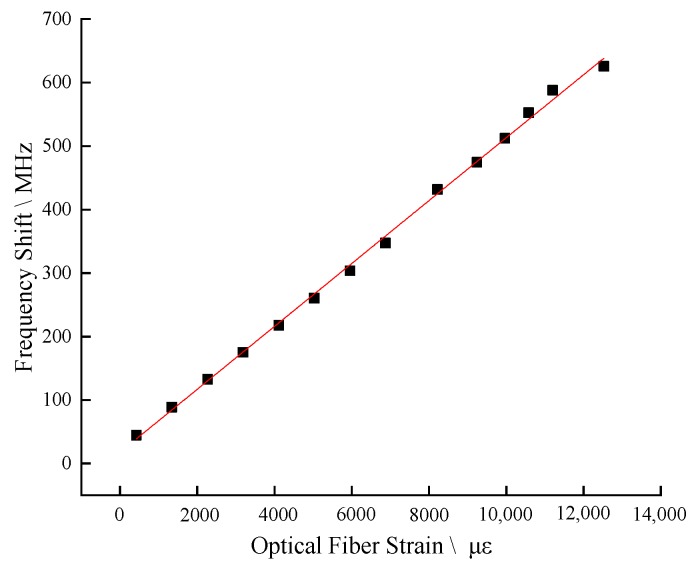
Strain coefficient of linear simulation.

**Figure 14 sensors-20-01318-f014:**
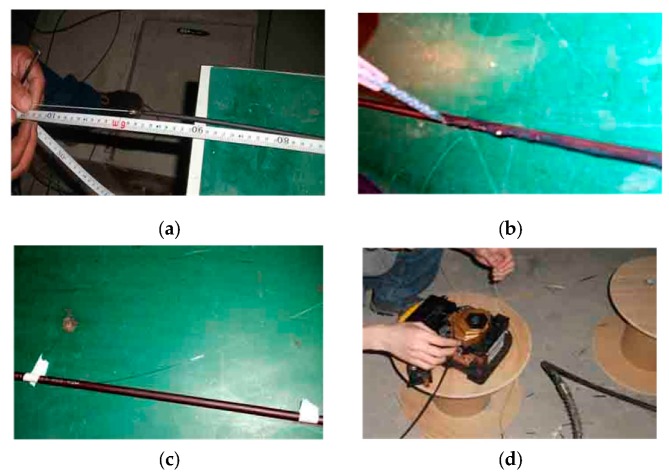
Preparation of test optical fiber. (**a**) Fiber length measurement. (**b**) Apply glue on the surface of SS fiber. (**c**) Adhere bare fiber to SS fiber. (**d**) Fusion of fiber and jumper.

**Figure 15 sensors-20-01318-f015:**
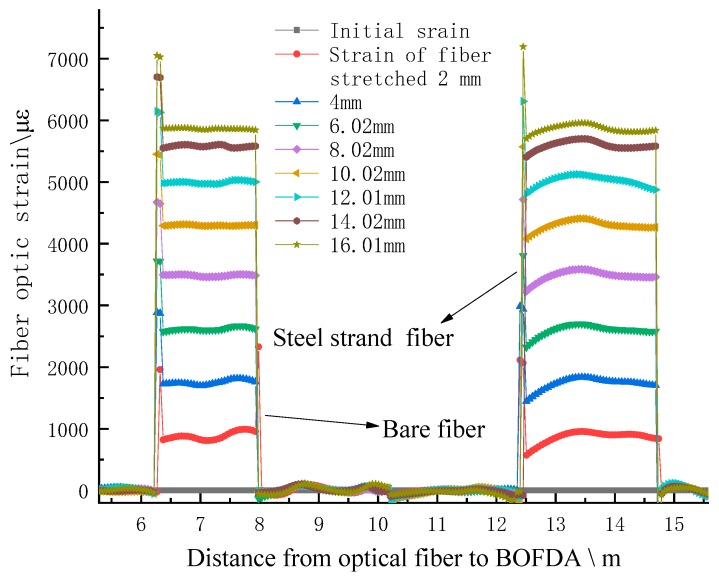
Strain curve of bare fiber and 5-mm steel strand fiber.

**Figure 16 sensors-20-01318-f016:**
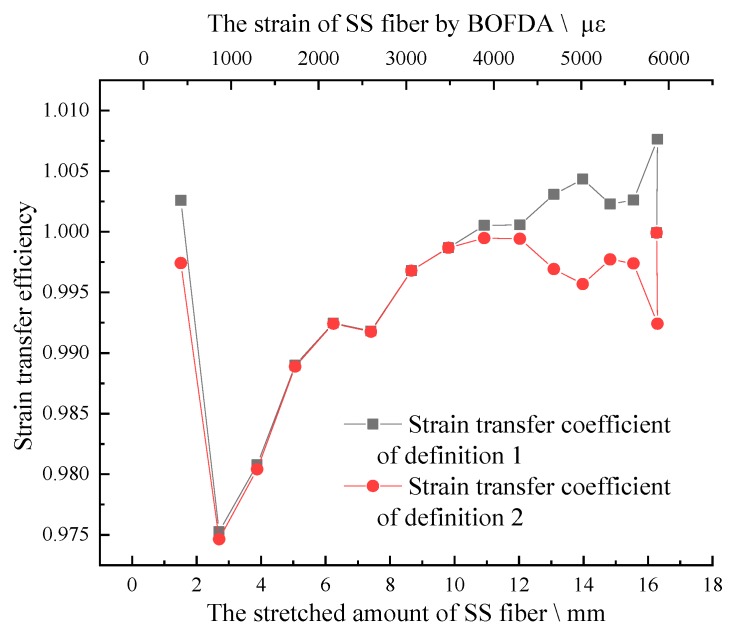
Strain transfer efficiency of SS fiber.

**Figure 17 sensors-20-01318-f017:**
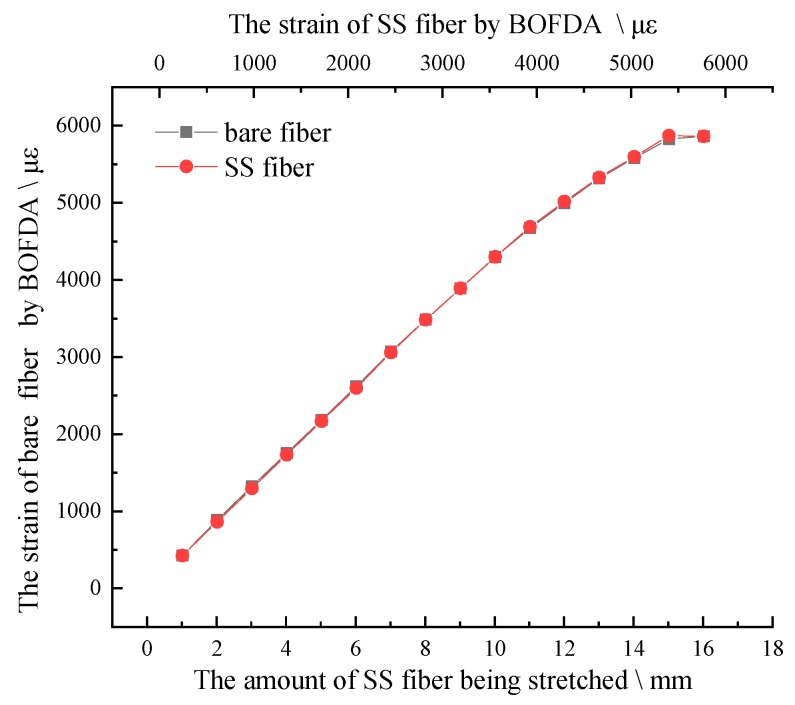
Comparison of strain curves between bare fiber and SS fiber.

**Figure 18 sensors-20-01318-f018:**
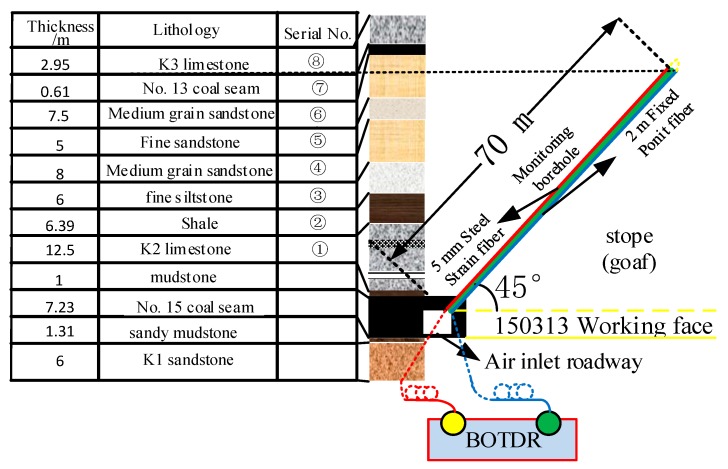
Optical fiber layout and roof/bottom rock formations.

**Figure 19 sensors-20-01318-f019:**
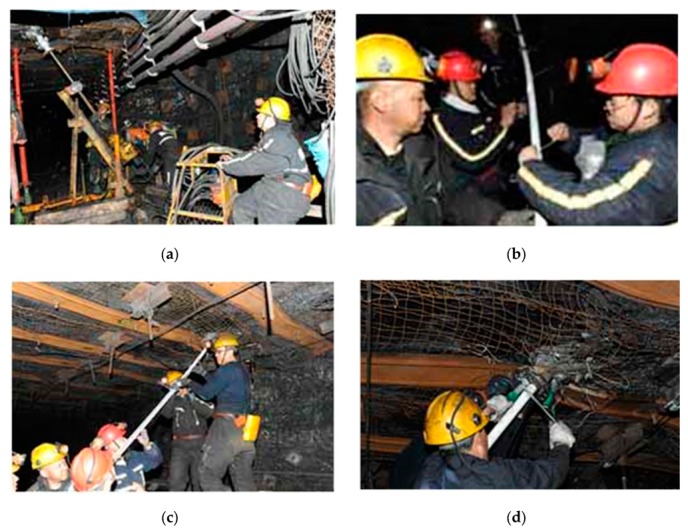
Process of embedding optical fiber in roof rock. (**a**) Drilling a borehole. (**b**) Fiber binding to the outer wall of PVC pipe. (**c**) Laying fiber through PVC pipe. (**d**) Grouting and sealing in borehole.

**Figure 20 sensors-20-01318-f020:**
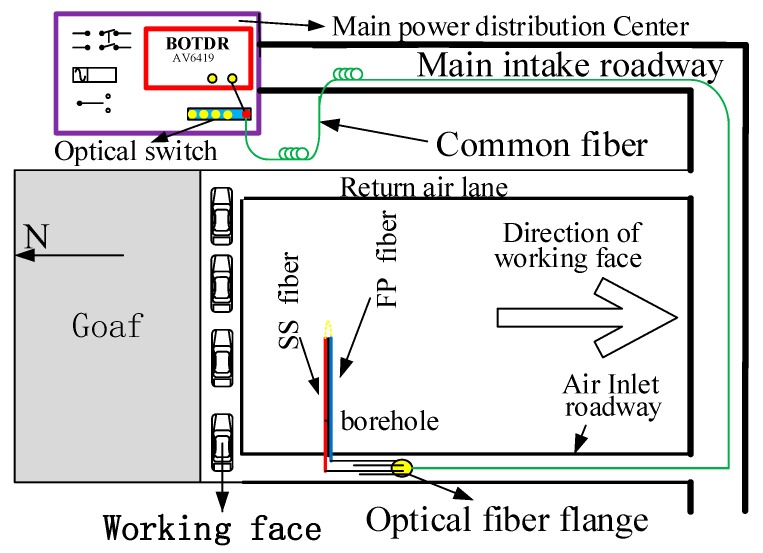
Roof strata movement monitoring system of BOTDR.

**Figure 21 sensors-20-01318-f021:**
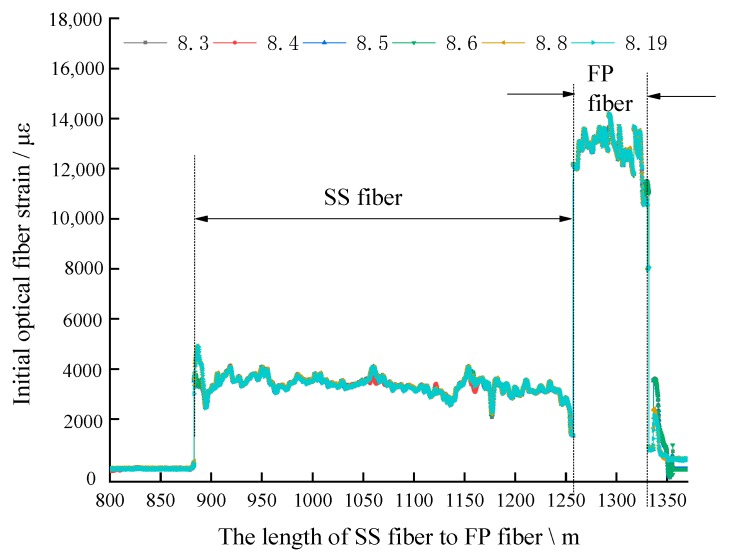
Original initial strain curve of optical fiber monitoring line ①.

**Figure 22 sensors-20-01318-f022:**
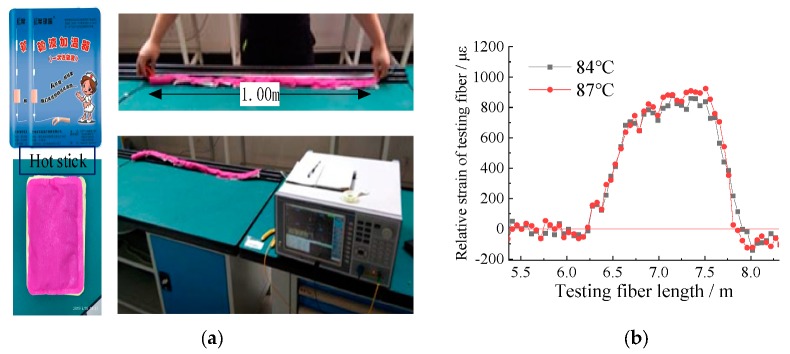
Optical fiber spatial position calibration test with BOTDR using hot deposition method. (**a**) shows the hot stick, test setup, and the test process; (**b**) shows the strain curve of the experiment.

**Figure 23 sensors-20-01318-f023:**
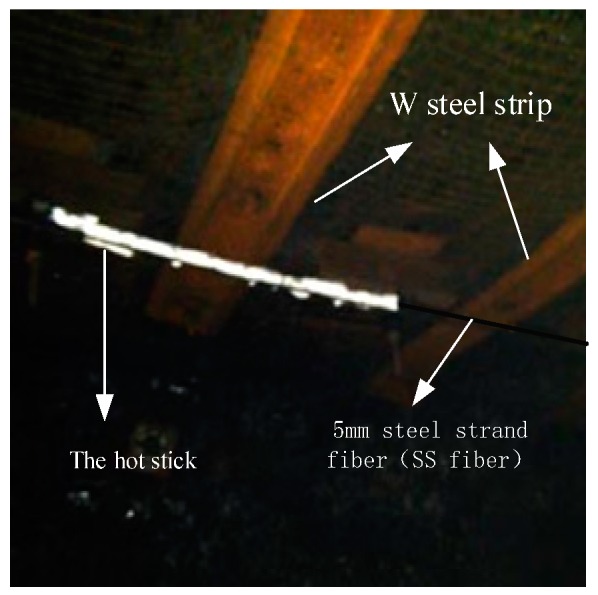
Location calibration of downhole optical fiber hot deposition method.

**Figure 24 sensors-20-01318-f024:**
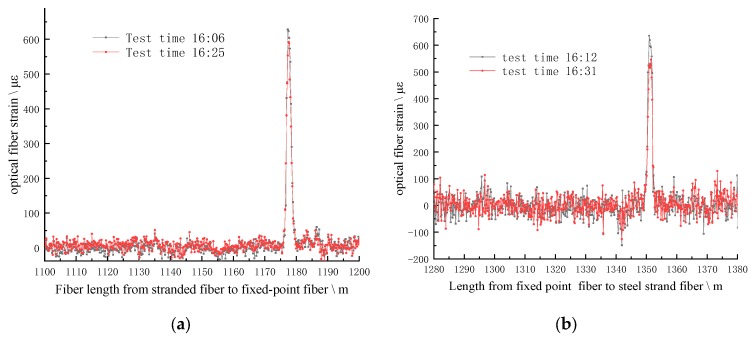
Spatial position calibration of optical fiber in coal mine. (**a**) shows the fiber strain curve of line ①; (**b**) shows the fiber strain curve of line ②.

**Figure 25 sensors-20-01318-f025:**
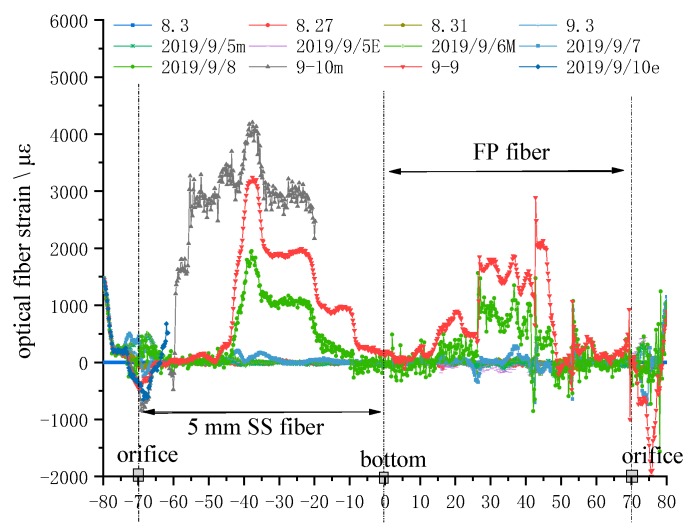
Strain of optical fiber from steel strand to fixed-point optical cable of line ①.

**Figure 26 sensors-20-01318-f026:**
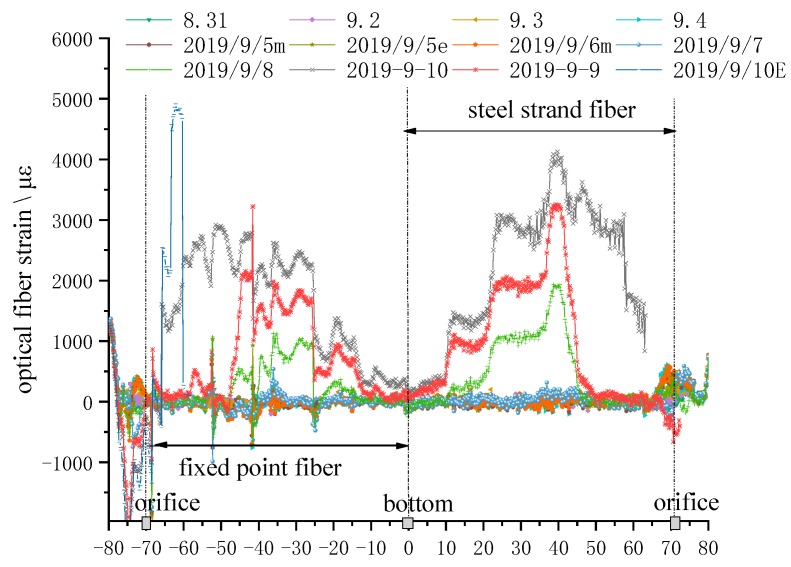
Strain curve of optical fiber of monitoring line ②.

**Figure 27 sensors-20-01318-f027:**
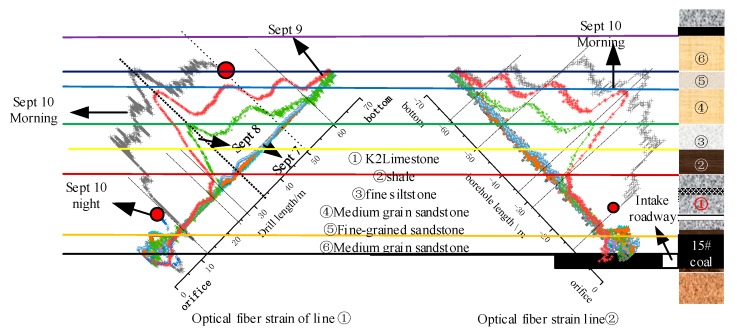
Stratigraphic comparison of strain curve of steel strand optical fiber in line ① and line ②.

**Figure 28 sensors-20-01318-f028:**
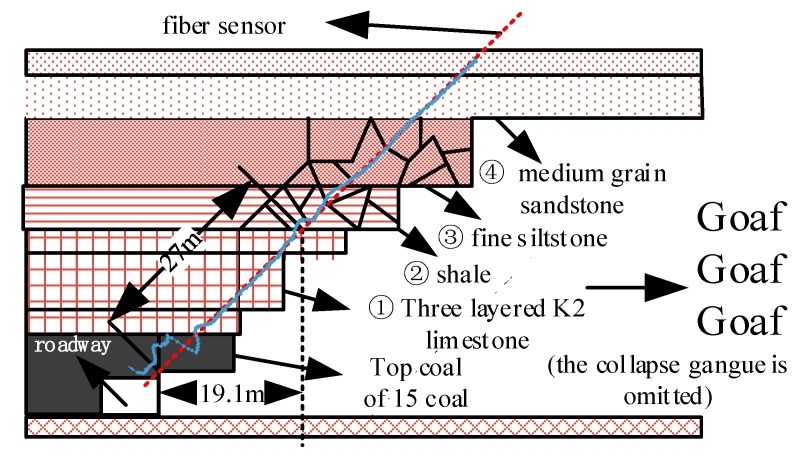
Schematic diagram of rock formation deformation on the fiber-optic section on 7 September.

**Figure 29 sensors-20-01318-f029:**
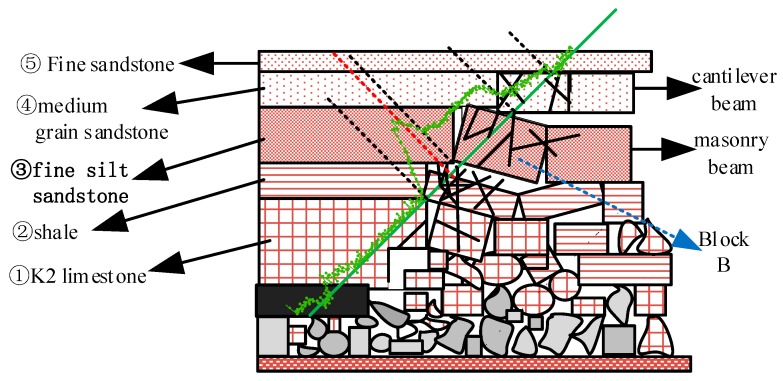
Schematic diagram of rock formation deformation on the fiber-optic section on 8 September.

**Figure 30 sensors-20-01318-f030:**
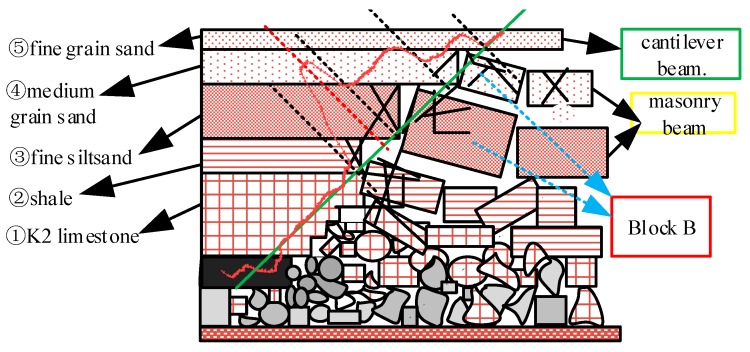
Schematic diagram of rock formation deformation on the fiber-optic section on 9 September.

**Figure 31 sensors-20-01318-f031:**
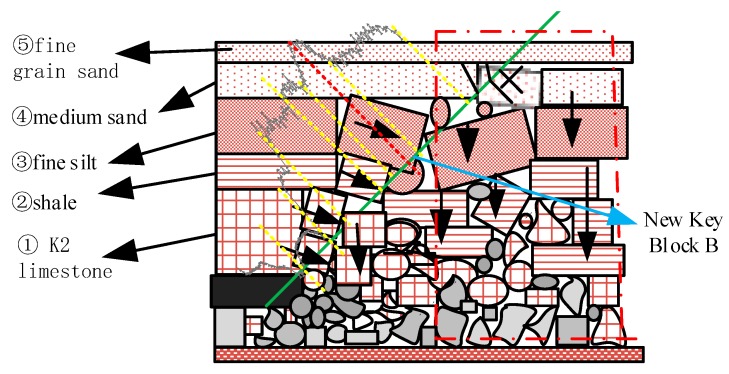
Schematic diagram of rock formation deformation on the fiber-optic section on 10 September.

**Table 1 sensors-20-01318-t001:** Specification of optical fibers selected in the test.

Fiber Type	Diameter/ mm	Tensile Strength/ MPa	Strain Range/ %	Unit Weight/ kg/km	Features
SS fiber	5.0	6.5	−2~2	42	High strength with high strain transfer performance
FP fiber	5.0	3.0	0~5	38	Continuous large deformation monitoring

**Table 2 sensors-20-01318-t002:** Specifications of the AV6419 analyzer.

Parameter Type	Specific Parameter
Fiber type	Single mode
Measurement range (km)	0.5, 1, 2, 5, 10, 20, 40, 80
The interval of spatial sampling (m)	0.05, 0.10, 0.20, 0.50, 1.00
Test pulse width (ns)	10, 20, 50, 100, 200
dynamic range (dB)	3.5, 7.5, 11.5, 14.5, 16.5
Repeatability of strain test (με)	≤±100
Strain test range (με)	−15,000–+15,000
Frequency scanning range (GHz)	9.9–12.0
Frequency scan interval (MHz)	1, 2, 5, 10, 20, 50
Working wavelength (nm)	1550 ± 5
Maximum number of sampling points	20,000

**Table 3 sensors-20-01318-t003:** Actual parameters used in field test of BOTDR.

Parameter Type	Specific Parameter
Measuring range/km	5
Spatial resolution/m	1
Sampling resolution/m	0.2
CS: MHz/με	0.0495
Initial Brillouin frequency/GHz	10.82
initial scanning frequency index/GHz	10.45
End scanning frequency/GHz	11.8
Optical fiber refractive index	1.468
Cumulative times	2^13^
Pulse width/ns	20
Frequency interval/MHz	10
Data points	20,000
Frequency points	136

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
