# Peer review of "The Field Monitoring Experiment of the Roof Strata Movement in Coal Mining Based on DFOS"

_sensors, 2020, doi:10.3390/s20051318_

Round 1

Reviewer 1 Report

The manuscript "Research on distributed optical fiber sensing technology of roof strata movement" describes an investigation on the use of distributed fiber optic sensors (DFOS) based on the Brillouin scattering effect in optical fiber for the assessment of mining operations. The paper starts with an overview of the state of the art of application of DFOS to geotechnical engineering in general and mining monitoring in particular. Then, the fundamentals of DFOS based on the Brillouin scattering effect are reviewed. The authors then move on to discuss the characterization and calibration of the optical fiber sensing cables deployed in the field using several laboratory test setups. Finally, the installation of the sensing cable on an operational mine is described followed by a discussion of the strain measurement results and their relation to the mining operation progress.

The manuscript describes a field-trial of a potentially interesting application of DFOS. The novelty of the work is regarded as average because there are previous works in the literature that describe similar field experiments. However, geotechnical engineering is a potential area of application of DFOS in which a great R&D effort is still needed to bring the solutions to practical use. Hence, the paper could be a significant contribution to the field. However, the big problem of the paper is the quality of the presentation and especially its very deficient use of the English language, which makes it difficult to follow the explanations and understand the content. This makes it impossible to properly rate the scientific contribution of the work described.

Therefore, it is the opinion of this reviewer that the manuscript is not suitable for publication in Sensors in its current state. The following major issues need to the addressed:

1. Thoroughly revise the use of English. Maybe the authors should consider a professional copy-editing service.

2. Re-write the abstract. A good abstract should provide a quick overview of the work highlighting its main contributions and conclusions. The current abstract of the manuscript enters into excessive details that are to be left for the main body of the manuscript. More general ideas are expected to let the reader grasp the importance of the work.

3. The style of the citations should be revised. Different, not-consistent citation styles are used throughout the text. It is advised that the authors review the journal guidelines and/or check papers previously published in this journal.

Author Response

Point 1:Thoroughly revise the use of English. Maybe the authors should consider a professional copy-editing service.

Response 1: According to the reviewer's opinion, 1. It has been modified by a native English engineer in South Africa who is my best friend and college when I worked in SA In 2001-2003.

Point 2:Re-write the abstract. A good abstract should provide a quick overview of the work highlighting its main contributions and conclusions. The current abstract of the manuscript enters into excessive details that are to be left for the main body of the manuscript. More general ideas are expected to let the reader grasp the importance of the work.

Response 2:  According to the opinions and suggestions of the reviewers, the abstract of this paper has been rewritten.

Point 3:The style of the citations should be revised. Different, not-consistent citation styles are used throughout the text. It is advised that the authors review the journal guidelines and/or check papers previously published in this journal.

Response 3: According to the reviewer's opinions and suggestions, the style of the citations have been revised by referring to the previous articles published in this journal sentence by sentence in detail. The citation style is mainly based on Google scholar’s GB/T7714

Reviewer 2 Report

In this paper, the distributed optical fiber strain monitoring technology is used to monitor the deformation and failure of roof strata in the process of mining, and gives the performances test results. I have the following comments:

Distributed optical fiber monitoring technology is very common for the measurement of temperature and strain. What is the biggest innovation of this paper or what is the difference or improvement with others?

What is the significance of the calibration of SS optical fiber strain sensing coefficient?And what is the guidance significance for the later application?

Some of the figures in the paper are not readable because of the small font size or the blurred pictures

Author Response

Response to Reviewer 2 Comments

Point 1:Distributed optical fiber monitoring technology is very common for the measurement of temperature and strain. What is the biggest innovation of this paper or what is the difference or improvement with others?

Response 1: Other scholars mainly use distributed optical fiber to monitor the height of caving zone and fracture zone caused by the deformation of mining overburden. Others researches on the movement law of overburden mainly focus on the indoor similar model test, while the reports of research on the dynamic field monitoring of overburden movement are fewer reported. In this paper, the field monitoring of overburden movement dynamic process is preliminarily tried by distributed optical fiber based on BOTDR

The biggest innovation is the whole process of roof strata from initial stable state to fracture failure and finally to re-stability is revealed based on BOTDR. This is helpful to optimize the support design in the coal mining and avoid roof accidents and so on.

The improvement and difference from other scholars' research is that this paper attempts to study the process and cause of the formation of the caving zone and fracture zone through the distributed optical fiber rather than the height.

Point 2What is the significance of the calibration of SS optical fiber strain sensing coefficient?And what is the guidance significance for the later application?

Response 2: The significance of SS optical fiber strain coefficient calibration is to determine the strain coefficient of BOTDR instrument in the field test. The calibration test of strain transfer efficiency of SS optical fiber is to confirm the percentage of strain transferred to optical fiber caused by the deformation of structure in the actual test. If the transfer efficiency is too small, the optical fiber is not suitable for field test; if the transfer efficiency is close to 1, it meets the requirements of field monitoring.

There are two guidance significance for the later application, the first is to make us sure that SS optical fiber meets the requirements and is competent for monitoring roof strata movement; the second is that the actual measured strain can be calibrated through the strain transfer coefficient when necessary. Because the transfer coefficient obtained from our indoor test is greater than 0.99, this paper considers that the strain loss can be ignored, so this test has not been calibrated.

Point 3Some of the figures in the paper are not readable because of the small font size or the blurred pictures.

Response 3: According to the reviewer's opinion, all the figures in the manuscript have been revised again, increasing the resolution and clarity of the figures.

Reviewer 3 Report

please check the attachment.

Author Response

Point 1:The title is so generalization, it covers too much research fields in DOFS monitoring for mining engineering. Also, it is without character, which could not express the main contributes and significance of this study. The abstract writing is poor and not appealing, please consider to reorganize it in the order of background, key questions, methods, discussion, and conclusions.

Response 1:

(1)According to the reviewer's comments and suggestions,The title of the manuscript is revised to:” The field monitoring experiment of the roof strata movement in coal mining based on DFOS”.

(2)The abstract is revised to:” Mining deformation of roof strata is the main cause of gas accidents, water inrush accidents and roof accidents in a coal mine underground. In order to ensure the safe mining of 150313 working face and effectively prevent the occurrence of accidents, Yinying Coal Mine adopts distributed optical fiber sensor technology to monitor the movement of roof strata in the goaf, so as to grasp the movement law of roof strata and make it serve for coal-mining. It is designed to drill boreholes in the roof angle of the coal wall of the roadway to the overburden of the stope, lay out optical fibers in the holes, and use the BOTDR to carry out the field monitoring test. For the success of the field test, the coupling test of the fiber strain in the concrete anchorage, the calibration test of the fiber strain coefficient of the 5 mm Steel Strand (SS) optical fiber and the laboratory test of the strain transfer performance of the SS fiber are carried out in the laboratory; the layout method of the fiber in the borehole and the calibration method of the spatial position of the fiber in the mine are carried out in the field. The indoor test results show that the high-strength SS optical fiber has a high strain transfer performance, which can be coupled with the concrete anchor under the action of external force and deform uniformly. The feasibility of SS optical fiber in the monitoring of rock strata movement is demonstrated theoretically and experimentally; the whole process activity law of roof strata fracture and collapse is obtained from the field test results. This paper is a trial test of the whole process of dynamic movement of roof strata mining deformation. It is hoped that the test results will help Yinying Coal Mine to optimize mining design efficiently and safely, prevent coal mine accidents, and provide detailed test basis and reference for optical fiber monitoring of mining deformation of coal seam roof underground”

Point 2:①There are a little lack of basic review of DOFS technology in the introduction, based on the main contents, it should review the mainstream commercial DOFS technology and analyze the effect of their application in field test and indoor experiment. Based on the introduction, ②it can be seen that the DOFS could turn the black box issue in mining engineering into a grey box issue even white box, but compare with traditional approaches, for example in the monitoring of water conduct zone height, what are the unique capabilities of DOFS ? At the same time, for the application environment of mining engineering, what opportunities and challenges will it face?

Response 2:

According to the comments of reviewer 1, I added and enriched the review of DFOS as bellows:

DOFSs  emerged  with  the  development  of  a  technique  called  Optical  Time- Domain Reflectometer (OTDR) in the early 80s. There are three main types of OTDR sensing mechanism, which are based on Brillouin [20], Rayleigh [21], and Raman [22] scattering.

The main disadvantage of Rayleigh-based DOFS is that the Rayleigh scattered signal may be weak, but they are sensitive to several physical parameters besides temperature and strain, namely, relative humidity [23], concentration of chemicals [24, 25], radiation detection [26], vibration and intrusion [27], among others. What is the main use is the Distributed Acoustic Sensing (DAS) as we known.

Raman distributed optical fiber temperature sensor system, which was reported in the  mid-80s,  is  one  of  the  most  successful  DOFS  developed  to  date. This sensor employs the Raman scattering in the optical fiber to measure temperature values. The Raman backward scattering is modulated by the temperature gradient along the fiber axis, making it possible to determine the real-time temperature distribution in the fiber [28], that is to say Distributed Temperature Sensing (DTS)

Brillouin-based distributed optical fiber sensors (Brillouin DFOS) have gained high academic and commercial interest due to their ability to provide distributed temperature and strain measurements along a several tens of kilometers long sensing fiber, with a high sensitivity and spatial resolution down to a few centimeters [29]. There exist practically two approaches to implement distributed Brillouin fiber sensors: (i) using spontaneous Brillouin scattering (SPBS) and (ii) using stimulated Brillouin scattering (SBS) [30]. Spontaneous  scattering  which  relies  on  the  detection  and  the  analysis  of  the backscattering  of  a  modulated  pump  signal;  depending  on  the  type  of modulation,  the  interrogator  is  commonly  called  Brillouin  Optical  Time  Domain Reflectometer  (BOTDR), Optical  Frequency  Domain  Reflectometer (BOFDR), Brillouin Optical Coherency Domain Reflectometer (BOCDR) etc.; Stimulated  Brillouin  Scattering  (SBS)  which  relies  on  the  detection  and  the analysis of a backscattered light wave  which is the product of the interaction of a pump and a probe signals; depending on the type of modulation, the interrogator  is  commonly  called  Brillouin  Optical  Time  Domain  Analyzer (BOTDA), Optical Frequency Domain Analyzer (BOFDA), Brillouin Optical Coherency Domain Analyzer (BOCDA) etc.[31].

The main stream DFOS widely used in strain monitoring based on Brillouin Scattering technology are the BOTDR, BOTDA and BOFDA. They have been applied to the strain monitoring of various kinds of structures in many industries, such as high anchored pile wall in Gravel [30], Bored Pile [31], Pile Foundation [32], Beam Bridges [33], similar material test model [34], Steel Rails[35], geotechnical structures and subsea cables[36,37], River Levee Collapse , Concrete Structures and yacht[38],tunnel healthy[39],the state of underground mine[40].land subsidence[41], Airplane structure health [42] etc.

②it can be seen that the DOFS could turn the black box issue in mining engineering into a grey box issue even white box, but compare with traditional approaches, for example in the monitoring of water conduct zone height, what are the unique capabilities of DOFS ?

Compared with traditional deformation monitoring methods, common advantages of every kind of optical fiber sensors come from its small size and light weight; low transmission loss; electrical isolation; security; flexibility; large bandwidth; reliability and low cost; robust; its non-electrical nature, making them immune to EM interference and to electrical noise, also allowing them to work into explosive environments; they usually have a very high sensitivity and a wide operating temperature range. Especially DFOS has the following advantages: ①high spatial resolution, high sensitive to strain of fibers;②it can realize distributed measurement. From one end of the optical fiber, the information of stress, temperature, vibration and damage at any point along the optical fiber can be obtained accurately, and these information are linear information, which can overcome the shortcomings of traditional point monitoring and improve the monitoring efficiency;③ the maximum range reaches 80 Km, which can meet the needs of large or super large structures, and the sensing optical fiber can be used as both the sensing body and the transmission body, which can realize long-distance and all-round monitoring.

For example in the monitoring of water conduct zone height, the unique capabilities of DOFS are its high spatial resolution, high sensitive to strain of fibers, light and flexible is easily lay out in different environments; distributed monitoring; long distance monitoring; and dynamic monitoring; especially its non-electrical nature, making them immune to EM interference and to electrical noise, also allowing them to work into explosive environments.

③At the same time, for the application environment of mining engineering, what opportunities and challenges will it face?

Through the discussion of the wide application and unique advantages of DFOS, DFOS can be applied to monitor the mining deformation of overburden in the indoor similar simulation test; to carry out the dynamic observation of the height of the water conducting fracture zone in the coal mine based on online real-time monitoring; to carry out the real-time monitoring of the health status of the underground roadway in the coal mine to prevent the occurrence of roof accidents; to carry out the real-time monitoring of the temperature of the goaf in the coal mine to prevent the occurrence of fire; to carry out the online real-time underground mine pressure observation to prevent the rock burst accidents., to real-time distributed monitor the underground water quantity In order to prevent the occurrence of water inrush accidents measures, and to real-time monitor the underground gas content variation by DFOS to prevent the occurrence of gas accidents and to online real-time monitoring the stability of stope and the subsidence of ground surface to protect the ecological environment,etc. Therefore, in the author’s opinion the biggest opportunity and challenges is how to fulfill the online real time monitoring in a coal mine production using DFOS. If so, it is the truly meaning of distributed sensing monitoring different from traditional monitoring methods. What we do using DFOS should be to monitor the process of overburden movement rather than the only result once a time.

Point 3: ①In section 2.1, it shall indicate which principle of DOFS technology is it belongs to. ②And the sensitivity of optical fiber sensing instruments is usually high, 5 degrees Celsius could has triggered a large strain or temperature testing errors, the compensation arrangement is very necessary and should not to be ignored The reason that temperature variation is not exceed 5 degrees Celsius to give up temperature compensation has no leg to stand on..

Response 3:

 ①The tittle of this passage 2.1 is revised to “principle of distributed optical fiber strain monitoring based on BOTDR and BOFDA”.

②About the temperature compensation, this part is revised to “Another outstanding advantage of Brillouin scattering light compared with other scattering light is that its frequency shift variation and temperature dependence are much smaller than that of strain (20µε/℃.). As the temperature has much less effect on the Brillouin frequency shift than the strain, the temperature change can be ignored or be discounted based on the temperature compensation [66]. A temperature sensor or a free optical fiber free from external force can be used for temperature compensation [67] Thus, in the indoor optical fiber test, the test period is very short and the temperature is basically constant. At this time, the influence of temperature on frequency drift can be ignored; In the underground field test, the free section optical fiber which is not affected by external force as the communication function is used to conduct temperature compensation for strain measurement. According to the relationship between the strain of optical fiber and the Brillouin frequency shift, the strain of optical fiber can be obtained according to formula (7)”

Point 4: In section 2.2, it is important to study the strain transfer performance of the optical fiber and the monitoring structure, but there seems to be something wrong with the experimental design in this experiment, the figure 4 to figure 6 demonstrated an obvious problem which is not really a problem. It is not surprised that the section 2.2 proved that the measurement results of SS fiber is better than the FP fiber, because discuss in essence, the FP fiber is designed to monitor the predictable large deformation, such as the ground fissure or fault, the SS fiber is designed to monitor the whole deformation of high strength structures, the two types of optical fiber serve different monitoring objects, in the concrete anchorage pull out test as shown in the paper, of course the SS fiber is indeed better than the FP fiber, and there is no more information to prove the SS fiber is better than other specially designed fibers.

Response 4:

From Fig.4 to Fig.6, it can be seen SS fiber couple bitter with the failure of concrete anchor solid than PP fiber clearly, and the most important is that the three strain changes of the fiber in the concrete anchorage can be used to accurately represent the movement state of the rock strata. ①When the fiber is in the initial strain state and the strain peak change is small, although the anchor solid may have slight strain change under the action of strata deformation, the damage of the anchor solid is small, and the surrounding rock strata where the anchor solid is located is in the original rock movement state  or in a state of strata movement as a whole.②When the peak strain curve appears in the optical fiber, the anchor solid failure at the peak position indicates that the position is in the separation position, and the strata of the position has been broken ,strata on two sides of the peak strain belong to different rock block, or it indicates that the lower rock stratum collapses, or it indicates the horizontal shear movement of the upper and lower rock strata, The peak strain of optical fiber indicates the degree of vertical separation of the rock or the degree of the horizontal displacement of the strata causes the horizontal shear failure of the anchor”.  All these analysis are added to this section of the manuscript.

Point 5: In section 2.3.2, the idea of experimental design is good, but it only verified the deformation consistency of SS fiber and bare fiber, and did not discuss the strain transfer model or strain transfer coefficient of the optical fiber and rock, author should to check out the strain transferring mechanism. The steel structure used in this experiment has a good deformation continuity, while the rock structure have complex deformation characteristics, the test results did not provide the strain transfer information of optical fiber- rock stratum relationship that the reviewer expected. In the meantime, the commercial optical fiber sensor itself has good deformation coordination, the emphasis should be on the study of strain transfer model of optical fiber and rock deformation.

Response 5:

Firstly I do agree with the reviewer’s comments on this section 2.3.2, the emphasis should be on the study of strain transfer model of optical fiber and rock deformation. So we added the contents below to the section.” As in the previous section, we have verified that the coupling process of SS optical fiber and concrete anchor solid is consistent through the coupling test of SS optical fiber and concrete anchor solid, and Figure 6 also shows that the strain changes of SS optical fiber and strain gauge are consistent. The point of view of this paper is that the concrete anchor solid is drilled into the rock stratum in the field, and the failure deformation of the anchor solid reflects the failure process of the rock stratum. That is to say, the SS optical fiber arranged in the anchor solid can reflect the deformation and failure of the rock stratum. It can be divided into two aspects. When the strain changes little in the strain section of the optical fiber, it indicates that the rock stratum where the optical fiber is located is in the overall deformation state due to the external force; when the optical fiber has a large peak strain curve, the rock stratum at the peak position of the optical fiber is in the state of separation or horizontal shear failure, and the rock stratum at both sides of the peak belongs to a whole rock block in different states, at the two rock blocks In a different state of motion.”

Point 6: In figure 14, please modify the labeling of the length of the optical fiber. It shows that the optical fiber is installed in the vertical formation direction. It should be revised and keep consistent with figure 22, so that the reader will not misunderstand the form and size of the embedded fiber in the rock stratum.

Response 6:

According to the suggestion and comment, the Fig.18 (Fig.14 before) has been revise to the actual layout, set at an angle of 45°, 70 m long with the correspondent lithology of different strata. It is consistent with Fig. 26(Fig.22 before)

Point 7: ①In section 4, the line 374 to line 378, it will be more professional to demonstrate the technique parameters in a table. ②In section 4.1.1, it is doubted that the maximum temperature and holding temperature of hot sticker, and it seems that the maximum temperature of commercial hot sticker will not exceed 50 degrees Celsius. ③Are there any other approaches could be better to realize the positioning of optical fiber? The spatial positioning should to be considered in the first stage of optical fiber installation, the hot sticker could only play the role of assistance verification, the reviewer thought.

Response 7:

①According to the suggestion of the reviewer, the technique parameters has been revised to a table. The detail can be seen in the revised paper.

② The Fig 21 has been changed. It is wrong with the Fig 21. A mistake was made about the temperature in data processing.  The temperature labelling is revised to 84 and 87℃

Question: Are there any other approaches could be better to realize the positioning of optical fiber?

Response: Also, We can use the different physical properties of various different optical fiber, use different strain transfer efficiency. When different fibers in the same circumstance, it will produce different strain curves to positon the spatial locaton. Those contents are added to section 4.1.1, also some relevant contents are added too, as below:

4.1.1. Spatial position calibration using initial strain in field test

As described in section 3.2 step1, the SS fiber and FP fiber are fusion to be a whole fiber as a loop in the borehole, it is defined as test line ① from the SS fiber to the FP optical fiber, and test line ② from the FP optical fiber to the SS optical fiber. According to the strain theory discussed above in section 2.1, the drilling position can be roughly positoned by the original strain curve, as shown in Fig. 21.

The original initial strain curve from the SS fiber to the FP fiber direction. It can be seen in the figure that 0-884 m is communication common optical fiber, 884-1257 m is 5 mm SS optical fiber, 1257-1332 m is FP optical fiber in the borehole, so it is impossible to accurately determine the position of SS optical fiber in the borehole; moreover, the positioning length of FP optical cable reaches 75 m, and the error between the actual borehole length of 70m is as much as 5 m. The error caused by this method of location based on the original strain curve makes the strain curve unable to accurately reflect the deformation movement of special roof strata in different lithology strata, let alone quantitative analysis. In order to achieve the accurate characterization of the law of movement of the strata with optical fiber strain, the accurate position calibration of the sensing optical fiber is the primary, most important and decisive work of roof strata movement monitoring.

The underground environment of coal mine belongs to the explosion-proof environment. The conventional chemical condensing agent quick freezing method, electric heating method and so on all violate the rules of coal mine production, so it is necessary to explore a new method for calibration.”

Point 8: From the context, it shows that there are 900 micro strain in line 395. Anyway, from the reviewer’s point of view, it seems that the author misunderstand the using of DOFS instrument. Under the influence of temperature, the readings on the instrument should be the Center Brillouin Frequency, or Brillouin Frequency Shift, then it could be conversed to strain or temperature by the strain/temperature sensitivity coefficients. Here, it seems improper that the changes triggered by temperature was displayed in strain. The same problem also shown in figure 17(b).

Response 8: The 900με is actually the relative strain indeed, which is the rise of strain. As shown in the figure below. This test was done when we selected methods from various materials to position spatial calibration in coal mine underground. Figure 2 is the experiment to test the relationship between the temperature variation and time of hot stick. Please see cover letter  of this reviewer in detail.

Point 9: In section 4.2,① what is the purpose of make two measurement loop for monitoring, one is FP fiber to SS fiber, and the other is in contrast.② In general, the main direction of rock stratum movement is accord with the gravity, and the axial strain is the main strain of optical fiber under pressure, so what is the basis of this paper to install the optical fiber into the ground at a 45 degree angle? ③Is there any consideration of expanding the DOFS results in vertical and horizontal directions to analyze the relationship of stratum movement and DOFS monitoring?

Response 9:

①The main purposes are: 1. To prevent the loss of data after the fracture of SS optical fiber in borehole by making use of the advantages of FP optical fiber, which is not easy to break due to large deformation; 2. To facilitate the observation channel, we hope to monitor the strain of SS optical fiber from both positive and negative channel directions, so as to improve the observation data of roof rock movement. The main advantages are: Even if the SS optical fiber breaks in the borehole, we can still obtain the strain curve of SS optical fiber from two directions to deduce the strain curve of the whole SS optical fiber in the borehole, and then study the movement of roof strata. ②The main reasons are as follows: 1. It is limited by the space of the roadway (in the middle is the frame for belt transportation, there is also the track for the tramcar to walk, the roadway section width is 5 m, and the section height is 3 m); 2. It is convenient to drill holes with a 45 degree angle (as required by the drilling team); 3. It is also more convenient for us to observe the deformation and movement of the roof strata in the horizontal and vertical directions through optical fiber strain, only a simple calculation formula can be used. ③Because the main study in this paper is to the research of rock strata in goaf, it is very important expanding the DOFS results in vertical and horizontal directions to analyze the relationship of stratum movement and DOFS monitoring. That is the manuscript should be revised to discuss and study in detail in below part of the manuscript.

Point 10: ①The annotation of Figure 20 and Figure 21 should be kept correspondence. ②Then, the strain measured by FP optical fiber are different with those measured by SS optical fiber in figure 20 and 21, so which of those results represented the real strain of rock stratum? ③Meanwhile, it seem like the results should be in same mode whatever the measurement loops it takes, but the figure 20 and figure 21 shown that they are not.

Response 10:

①     The annotation of Fig. 25 and Fig. 26 has been revised in consistence.(before Fig.20 and Fig.21)

②  In the manuscript, the writer’s viewpoint is that all the SS fiber strain results can represent the real strain of rock strata. But only the result of SS fiber strain part in line② from FP fiber to SS fiber can completely and comprehensively represent the whole process of strain deformation of rock stratum. Because in the morning of Sept.10 the SS fiber was broken in the borehole, only a few of strain data can be obtained; at the same period, line② from FP fiber to SS fiber can test most length of the whole fiber in the borehole, there were much more strain data obtained than line①.

③  The biggest different is that there is a part of SS fiber strain data in figure 24 because SS fiber was broken in the morning of Sept 10(the black dotted line shown); there are a whole fiber strain data  except at the orifice just as shown in the figures below.

Point 11:① In section 5, the paper demonstrates a good results of the roof rock stratum movement combining with the optical fiber in figure 23 to figure 26. But the strain distribution of DOFS monitoring were missed in those figures, which leads that the roof stratum movement cannot connect with the DOFS monitoring. ②Another important problem is that the mechanic analysis of rock block movement have no other proof, other than some theoretical hypothesis from the classic theories. The field engineering environment is harsh to investigate, there should be more monitoring or measurement results by other approaches to prove the roof stratum movement law, ③besides the DOFS monitoring. And the DOFS monitoring results cannot prove the movement law, such as, the cantilever beam state or other structural state cannot concluded only by the DOFS monitoring, or no other characteristic form of monitoring data curve or strain eigenvalue to prove those structural states. The contrast verification is in urgent need in those engineering applications.

Response 11:

①If the strain distribution is displayed in the drawing, it will be too crowded and too small to be read. According to suggestions of the reviewer, the actual strain curve has been added to the relevant figures. Show as in revised vision of manuscript.  ②The mechanic analysis of rock block movement is the integrated synthesis of Qian’s three important theory: the Voussoir Beam, the Key Block, the Key Layer and the “O shape” circle. Because Qian’s theory are obtained from simulation experiment and phenomena of real coal mining, this paper is a trial for confirming Qian’s theory through field monitoring of rock strata movement. There is few any other field test of strata movement based on Qian’s theory, from the view of writer of this paper, this kind of experiment is suitable for DFOS just like the perceptual neural network is deployed in the rock strata to sense any movement in the embedded rock strata. And to the writer’s knowledge of monitoring the progress of overburden mining movement, there is no other better method that with high resolution, high sensitivity, distributed, real-time, water proof, electric-magnetic isolation etc. in the inner of rock strata. ③The Voussoir beam and cantilever beam structure can be referred as Jinfeng JU [1] ,shown as below figure.I agree with the reviewer’s comments that it is impossible to determine the cantilever beam and masonry beam structure of the rock stratum only from the perspective of optical fiber strain. However, in the special case and conditions of monitoring the movement of roof strata in goaf, it can be determined by combining the physical knowledge of material mechanics, the pull-out test of concrete anchor and the strata lithology. Take fig. 30 as an example, the optical fiber in zone (a) is in initial strain state, and the rock stratum in this zone is in stable state; the optical fiber in the zone (b) is a curve with larger peak value, and the pull-out test can determine that the rock stratum in the area collapsed by the gravity of the rock stratum, and the part of rock stratum within the optical fiber is a larger separation position; The platform shaped strain curve appears in the zone (c), which indicates that the rock stratum is in the overall tension state. The only explanation is that the rock stratum rotates and sinks as a whole at this time. It shows that the displacement of the rock stratum as a whole in the zone (c) is greater than that in the zone (d) with the same characteristics, and there is a certain separation between the two layers of the rock stratum, The rock stratum in the zone (d) is also in the state of overall tension and overall rotation and subsidence; As the lithology of zone (c) and zone (d) are hard and brittle sandstone, if it is cantilever structure, the collapse of the rock layer will cause the peak curve of the embedded optical fiber, the results is that there is no peak curve, but there is a platform curve with small strain, which indicates that it is supported by the underlying collapsed rock layer, and it is obviously a temporary structure with Key Blocks occluding and supporting each other The rock layer of the optical fiber part is obviously the masonry beam structure defined by Qian Minggao; The rock stratum in the zone (e) is in a stable state, while the underlying rock stratum has rotated and sunk. It is obvious that the rock stratum in the zone (e) belongs to cantilever beam structure.

Point 12: In section 6, ①please carefully refine the conclusions, and express the main conclusion and innovation of this paper by simple conclude, it is unprofessional and nonacademic that take all the technical details as conclusions.② In addition, there will be more work to be down about the research mentioned in the paper, the statement in conclusion (7) is poor and inappropriate.

Response 12:

Firstly, I accept the reviewer’s comments, it is not appropriate to get the conclusion (7), this conclusion is deleted.

According to the reviewer’s comments, the conclusion is refined.

It is right there will be more work to be done, if the whole movement law of roof is to be mastered. The manuscript is only a trail experiment in order to explore the movement of strata. The writer hopes to do some contributions in the exploration from “black box” to “white box “in coal mining ,to broaden the application of DFOS and provide detailed test basis and reference for optical fiber monitoring of mining deformation of coal seam roof underground”

Point 13:It is commendable that this paper made a lot of basic calibration test in laboratory works before conducting field application. ①But it also has a weakness that the study only choose two optical fibers of FP and SS fiber into tests, there should be more comprehensive comparison researches of different structures, encapsulations and sizes of optical fiber sensor. ②Besides, the annotations or labeling of all figures in the paper need to be optimized, ③the English writing also need to be extensively checked. Because of those reasons above, the reviewer recommends that the paper should have major revision before publication consideration.

Response 13:

The reason why the manuscript only select to kinds of fibers is the two kinds fiber are mainly used in field study of water conducting fractured zone before, reference as [51-55] The reviewer’s comments are very correct, there should be more kinds selected in this manuscript’s field test, I agree. According to the comments, I re optimized most of the figures which are not clear and inappropriate about labeling and annotations. My native English friend ,Mr. Mike Inman in South Africa, has helped me to check the English writing and help me revised some English errors in writing, helped me decorated the English style overall. Mr. Inman is an engineer of coal mining, he does not understand well about DFOS, Maybe there might be some to be improved and optimized later, but I have do my best.  

Reference in the response.(the figrure can not shown in my response, in can be shown in coverletter of this reviewer)

[1]    Ju J, Xu J. Structural characteristics of key strata and strata behaviour of a fully mechanized longwall face with 7.0 m height chocks [J]. International Journal of Rock Mechanics and Mining Sciences, 2013, 58: 46-54.

Round 2

Reviewer 1 Report

The authors have performed a thorough revision of the manuscript and now it can be suitable for publication in Sensors provided the following issues are addressed:

1. The use of the English language has improved quite a lot, but there are some expressions that still need revising. Furthermore, there are a number of typos and punctuation errors that need to be corrected so that the paper achieves the word-class presentation quality required for the Sensors journal.

2. The authors need to bear in mind that they are writing a scientific paper, and scientific papers must have accuracy, clarity, and brevity. In the case of this manuscript, there are some parts in which these virtues are missing, particularly, brevity, which means saying only what needs to be said, avoiding wordiness and redundancy.

The abstract is somewhat wordy. The author should review it and think if it can be summarized to reflect the motivation and the state of the art related to their work.

The intro to DFOS technology in Section 2 is also excessively long. The authors should give only a very general idea of the working principle of the sensors they are deploying and refer the reader to references that properly introduce the technology. For instance, a good intro to DFOS is in Hartog's book "An introduction to distributed optical fibre sensors", but there are others.

It is also the opinion of this reviewer that current section 2 should be divided into two sections: a brief intro to DFOS (section 2) and another high-level section (section 3?) describing all the sensors' characterization and calibration work.

3. The authors should clearly explain the so-called fixed-point fiber (FP fiber) configuration, describing what type of fiber cable they are using and how it is anchored to the structure.

4. It would be interesting to have an additional figure to accompany the current fig. 7 that depicts strain in the middle part of the FP fiber vs. deformation.

5. Fig. 16 and Fig. 17 should have an extra x-axis in terms of strain in the SS fiber so as to make it possible to determine the linear range of strain transfer.

Author Response

Response to Reviewer 1 Comments

(2020/2/7)

Point 1:  The use of the English language has improved quite a lot, but there are some expressions that still need revising. Furthermore, there are a number of typos and punctuation errors that need to be corrected so that the paper achieves the word-class presentation quality required for the Sensors journal.

 Response 1:  According to the comments of the reviewer, this manuscript has been checked several times by the authors, some typos and punctuation errors have been corrected.

Point 2: The authors need to bear in mind that they are writing a scientific paper, and scientific papers must have accuracy, clarity, and brevity. In the case of this manuscript, there are some parts in which these virtues are missing, particularly, brevity, which means saying only what needs to be said, avoiding wordiness and redundancy.

(1)The abstract is somewhat wordy. The author should review it and think if it can be summarized to reflect the motivation and the state of the art related to their work.

(2)The intro to DFOS technology in Section 2 is also excessively long. The authors should give only a very general idea of the working principle of the sensors they are deploying and refer the reader to references that properly introduce the technology. For instance, a good intro to DFOS is in Hartog's book "An introduction to distributed optical fibre sensors", but there are others.

(3)It is also the opinion of this reviewer that current section 2 should be divided into two sections: a brief intro to DFOS (section 2) and another high-level section (section 3?) describing all the sensors' characterization and calibration work.

Response 2:

(1) The abstract has been revised again, the new abstract is as below:

       “Mining deformation of roof strata is the main cause of methane explosion, water inrush and roof collapse accidents amid coal mining underground. To ensure the safety of coal mining, the distributed optical fiber sensor (DFOS) technology has been applied in the 150313 working face by Yinying Coal Mine in Shanxi Province, north China to monitor the roof strata movement, so as to grasp the movement law of roof strata and make it serve for production. The optical fibers are laid out in the holes drilled through the overlying roof strata in the roadway and BOTDR technique is utilized to carry out the on-site monitoring. Prior to the on-site test, the coupling test of the fiber strain in the concrete anchorage, the calibration test of the fiber strain coefficient of the 5mm steel strand (SS) fiber and the test of the strain transfer performance of the SS fiber have been carried out in the laboratory; The approaches for fiber laying-out in the holes and fiber’s spatial positioning underground the coal mine have been optimized in the field. The indoor test results show that the high-strength SS optical fiber has a high strain transfer performance, which can be coupled with the concrete anchor with uniform deformation. This demonstrated the feasibility of SS fiber for monitoring strata movement theoretically and experimentally; and the law of roof strata fracturing and collapse is obtained from the field test results. This paper is a trial to study the whole process of dynamic movement of the deformation of roof strata. Eventually the study results will help Yinying Coal Mine to optimize mining design, prevent coal mine accidents, and provide detailed test basis for DFOS monitoring technique of roof strata movement”.  

(2) According to the reviewer’s comments, the intro of DFOS has been revised again. Mainly by referring to and quoting Hartog's books, DFOS is redefined as follows:

       “A distributed optical fibre sensor is defined as an intrinsic sensor that is able to determine the spatial distribution of one or more measurands at each and every point along a sensing fibre. DOFSs use optical fibres both as the sensing element and as the means of carrying the optical signals used for this purpose [55]”.

 (3) The authors agree with the reviewer very much, and divide this chapter into two independent chapters, as below:

       “2. Distributed fiber optical sensor (DFOS) technology

2.1. The principle of BOTDR and BOFDA

2.1.1 BOTDR

2.1.2 BOFDA

2.2 Principle of optical fiber strain monitoring”

         “3. Indoor experiments of fiber strain based on DFOS

3.1. Experimental study on strain transfer of optical fiber in borehole anchorage

3.1.1. Experiment method and experiment system setup

3.1.2. Analysis of experiment results

3.2. Strain transfer performance test of SS optical fiber

3.2.1 The introduction of the SS fiber and FP fiber

3.2.2. Calibration of SS fiber strain coefficient

3.2.3. Experimental on strain transfer performance of SS fiber

3.3. Introduction to the instruments selected for field experiment”

Point 3: The authors should clearly explain the so-called fixed-point fiber (FP fiber) configuration, describing what type of fiber cable they are using and how it is anchored to the structure.

Response 3: According to the reviewer’s comments, the detailed introduction of FP fiber is added to the Section 3.2.1, its configuration and structure is introduced too as bellows:

“The so-called FP fiber, that is, by artificially applying fixed points on the sensing fiber, and the deformation between two adjacent fixed points can be sensed by the fiber. The strain of the sensing fiber between the two fixed points is not affected by the deformation of the surrounding environment of the fiber, but depends on the relative displacement between the two fixed points. When the FP fiber is laid, it is usually just to stick or fix the aluminium alloy fixed tubes on the surface of the structure under test shown as Fig.10 (B) .This kind of fiber can solve the problems of fixed point method in monitoring large deformation of rock mass”.

(B) Structure and physical diagram of fixed-points optical fiber (FP fiber)

Fig. 10. Structure and physical figure of the SS fiber and the FP fiber

Point 4: It would be interesting to have an additional figure to accompany the current fig. 7 that depicts strain in the middle part of the FP fiber vs. deformation

Response 4: according to the comments of the reviewer, an actual picture of the anchor drown deformation adhered to the figure of FP fiber strain curve, shown as below and in the revised manuscript too.

(a)     The strain curve of FP fiber

(b)     The actual deformation of concrete column

Fig. 7. Strain curve of FP fiber (a) compared with the actual anchor drawn deformation.

Point 5: Fig. 16 and Fig. 17 should have an extra x-axis in terms of strain in the SS fiber so as to make it possible to determine the linear range of strain transfer.

Response 5: According to the reviewer’s comments, all an extra X-axis in terms of SS fiber strain have been added to Fig.16 and Fig.17. These can be seen in the revised manuscript, shown as bellows:

Fig. 16. Strain transfer efficiency of SS fiber.

Fig. 17. Comparison of strain curves between bare fiber and SS fiber.

Reviewer 2 Report

The author has revised the manuscript accordingly. It is recommended to accept the manuscript for publication.

Round 3

Reviewer 1 Report

The paper is suitable for publication.